# Drebrin controls scar formation and astrocyte reactivity upon traumatic brain injury by regulating membrane trafficking

Juliane Schiweck [1,4], Kai Murk [1,4✉], Julia Ledderose[1], Agnieszka Münster-Wandowski[2], Marta Ornaghi[1], Imre Vida [2] & Britta J. Eickholt [1,3✉]

The brain of mammals lacks a significant ability to regenerate neurons and is thus particularly vulnerable. To protect the brain from injury and disease, damage control by astrocytes through astrogliosis and scar formation is vital. Here, we show that brain injury in mice triggers an immediate upregulation of the actin-binding protein Drebrin (DBN) in astrocytes, which is essential for scar formation and maintenance of astrocyte reactivity. In turn, DBN loss leads to defective astrocyte scar formation and excessive neurodegeneration following brain injuries. At the cellular level, we show that DBN switches actin homeostasis from ARP2/3-dependent arrays to microtubule-compatible scaffolds, facilitating the formation of RAB8-positive membrane tubules. This injury-specific RAB8 membrane compartment serves as hub for the trafficking of surface proteins involved in astrogliosis and adhesion mediators, such as β1-integrin. Our work shows that DBN-mediated membrane trafficking in astrocytes is an important neuroprotective mechanism following traumatic brain injury in mice.

[1] Institute of Biochemistry, Charité - Universitätsmedizin Berlin, Berlin, Germany. [2] Institute of Anatomy, Charité - Universitätsmedizin Berlin, Berlin, Germany. [3] NeuroCure - Cluster of Excellence, Charité - Universitätsmedizin Berlin, Berlin, Germany. [4]These authors contributed equally: Juliane Schiweck, Kai Murk. ✉email: kai.murk@charite.de; britta.eickholt@charite.de

The brain of higher mammals is vulnerable to traumatic injury and disease, but largely lacks the capacity to regenerate neurons and rarely regains function in damaged areas[1]. This makes the containment of local pathological incidents critical to avoid the propagation of inflammation and neurodegeneration into the uninjured brain parenchyma.

Astrocytes are key players in the protection of CNS tissue via a defense mechanism known as reactive astrogliosis. This process is associated with comprehensive changes in astrocyte morphology and function[2]. Reactive astrocytes develop hypertrophies of soma and protrusions, while cells in proximity to large lesion sites polarize and extend particularly long processes. Dense arrays of such "palisades" and hypertrophic astrocytes constitute scars, which enclose as physical barriers inflammatory cues and extravasating leukocytes, and limit the spread of damage[3,4]. Astrocyte scars are well described anatomically and with respect to certain signaling events[5], but little is known about molecular mechanisms controlling the cytoskeletal organization during polarization and outgrowth of palisade-like astrocyte processes.

In the context of astrogliosis and scar formation, we studied drebrin (DBN), a cytoskeletal regulator, which stabilizes actin filaments by sidewise binding and by competing off other actin-binding proteins[6,7]. It is widely expressed, but has mainly been studied in neurons[8]. In cultured astrocytes, DBN has been described to maintain connexin 43 at the plasma membrane[9]. Functional coupling of astrocytes into networks through gap junctions is essential to modulate neuronal transmission[10,11]. A deficit in DBN would therefore be expected to cause profound phenotypes in vivo. However, mouse models with acute or chronic DBN loss indicate non-essential functions of this actin-binding protein during neuronal development, as well as during synaptic transmission and plasticity[12]. Instead, we identified DBN as important local safeguard mechanism in dendritic spines during conditions associated with increased oxidative stress[13]. Here, we characterize DBN as an injury-specific actin regulator in astrocytes, crucial for the wounding response and tissue protection in the brain. During injury, DBN provides a key switch to alter actin network homeostasis, which prepares the foundation for tubular endosomes, enabling polarized membrane trafficking of crucial surface receptors. In this role, DBN controls reactive astrogliosis required to form astrocyte scars and to protect the susceptible CNS from traumatic brain injury.

## Results

### DBN is an injury-induced protein in reactive astrocytes. To investigate endogenous DBN protein expression in astrocytes, we performed immunocytochemistry in 21 days in vitro (DIV) cortical cultures from mice containing both neurons and astrocytes. Although we did not detect any DBN signals in astrocytes, high DBN levels were found in dendritic spines of neighboring neurons (Fig. 1A). We exploited an in vitro scratch-wound model to induce astrocyte reactivity, which increased DBN expression with prominent labeling along the astrocyte processes (Fig. 1A). In purified astrocyte cultures, where cells adopt polygonal morphologies, antibody labeling, and western blot analyses identified low DBN expression (Fig. 1B, S1A). Following mechanical scratch injury, DBN protein levels were increased, reaching 3.2-fold of the baseline level 24 h post injury (Fig. 1B).

In the mouse brain, DBN was distributed in distinct puncta in proximity to MAP2-positive dendrites, but not in S100ß+ astrocytes (Figures S1B). To induce reactive astrogliosis, we used an in vivo model for CNS trauma and inserted a needle unilaterally into the cortex of BAC Aldh1l1 eGFP reporter mice[14]. Seven days after this "stab injury", eGFP expression allowed us to visualize astrocytes independently of their quiescent or reactive

status, while GFAP-labeled reactive astrocytes could be identified within the forming scar around the lesion site. After this stab injury, numerous injury-induced GFAP+ astrocytes exhibited DBN protein expression throughout their polarizing processes (Fig. 1, S1B, Movie 1). In summary, our results demonstrate that CNS injury triggers an immediate upregulation of DBN in astrocytes in vitro and in vivo.

### Reactive astrocytes require DBN to form glia scars in vivo. To investigate if injury-induced upregulation of DBN protein is required during astrogliosis responses in vivo, we looked at its functional significance in DBN knockout mice during scar formation following cortical stab-injuries[12]. At 7 days post injury, the stab wound triggered the formation of long palisading processes extending toward the injury site in 45% of GFAP-positive astrocytes in WT brains, in line with previous studies[3,15,16]. In contrast, in $Dbn^{-/-}$ mice, GFAP-positive astrocytes showed strong defects in polarization and in the formation of palisade-like barriers (Fig. 2A).

Next, we addressed whether the defect in scarring induced by DBN-loss affects the surrounding tissue. Microglia are highly sensitive sentinels in the brain, which become hypertrophic phagocytic cells after injury[17]. Although cultured microglia, as well as microglia in intact and injured brains do not express DBN (Figures S1C, D), they respond to injury with increased hypertrophy in $Dbn^{-/-}$ brains when compared with WT brains (Fig. 2B). This result indicates that stab injury in $Dbn^{-/-}$ brains exacerbate microglial responses, which likely reflects defective scarring.

We then asked if this defective scar development affects surrounding neurons. Loss of NeuN indicates degenerating neurons[18], and its translocation from the nucleus to the cytosol identifies neurons exhibiting an initial stress response to pathologies[19–22]. Following stab injury, WT brains consistently maintained NeuN in the neuronal nuclei irrespective of their position relative to the injury site. In contrast, in $Dbn^{-/-}$ brains, NeuN translocation from the nucleus to the cytosol was present in 42% of neurons (Fig. 2C). As our disease model is considered as mild injury, we conclude that $Dbn^{-/-}$ brains exhibit signs of increased brain damage 7 days post injury.

### DBN provides long-term tissue protection after traumatic brain injury. To study the long-term outcome of DBN deficiency, we extended the stab injury analyses to a later time point. At 30 days post injury, WT mice maintained well-defined GFAP+ scars exhibiting typical signs of long-term thinning (Fig. 2D)[15], whilst we no longer detected GFAP+ astrocytes in lesion sites of $Dbn^{-/-}$ brains (Fig. 2D, Figure S2A). To verify these findings, we also used vimentin as additional marker for astrocyte reactivity. Analogous to GFAP, WT astrocytes showed prominent vimentin levels in scars 30 days post injury, while $Dbn^{-/-}$ astrocytes were negative for this marker protein (Figure S2B). To analyze the fate of astrocytes in lesions without GFAP and vimentin immunoreactivity, we stained for SOX9, a transcription factor and nuclear marker for mature astrocytes[23]. In WT brains, we consistently detected Sox9-positive nuclei in astrocytes within scars. In contrast, in core lesions of $Dbn^{-/-}$ brains, SOX9 exhibited a strictly cytoplasmic localization (Fig. 2D), as reported for undifferentiated stem cells and precursor cells[24]. This cytoplasmic localization was scar-specific, as in the uninjured tissue adjacent to stab wounds in $Dbn^{-/-}$ brains, SOX9 showed its typical nuclear localization (Figure S3A). Cytoplasmic SOX9 in the scar tissue correlated with the absence of GFAP in all injured $Dbn^{-/-}$ brains, which led us to hypothesize that $Dbn^{-/-}$ astrocytes at the lesion sites are present, but do not maintain their typical reactivity.

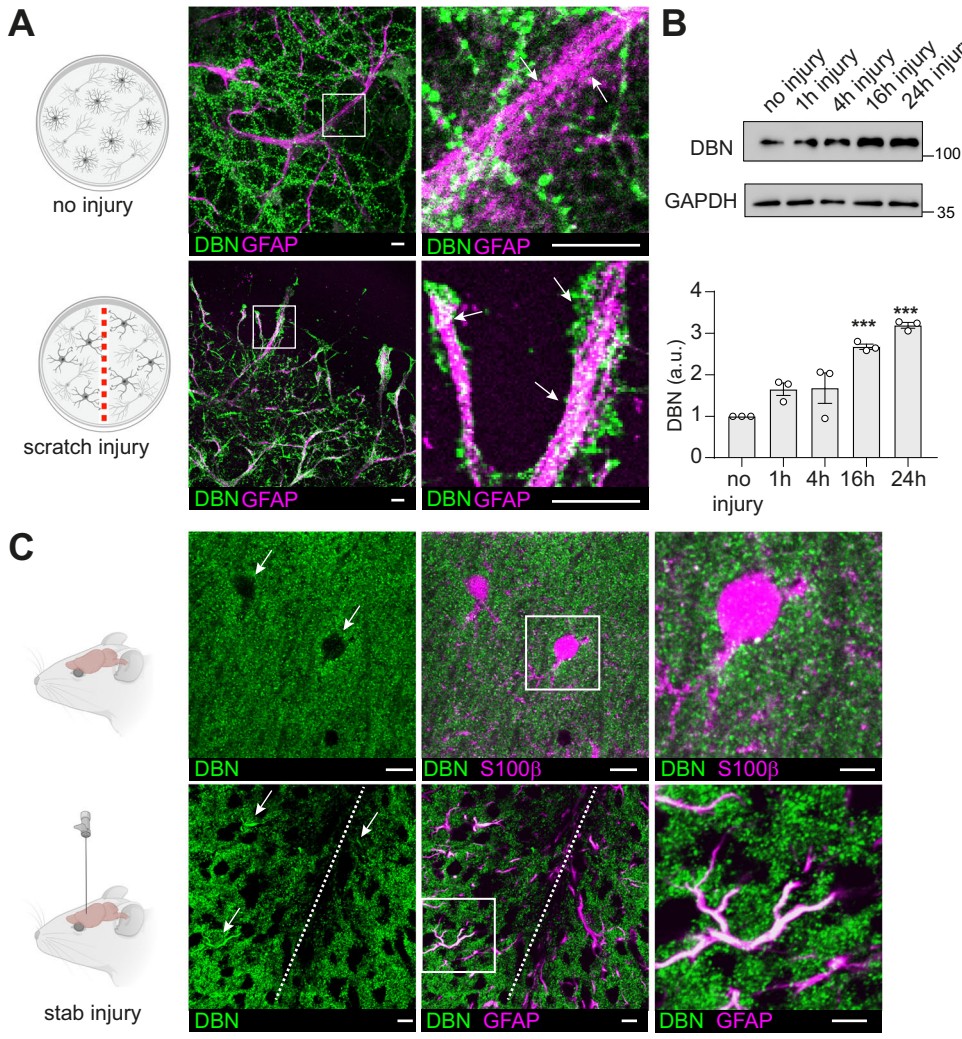

**Fig. 1 DBN protein levels are increased in response to injury. A** DBN labeling (green) in mixed 21 DIV cortical cultures without (upper panels) or with mechanical scratch injury (lower panels). Square in the left image is magnified on the right. Arrows show GFAP+ astrocyte processes, devoid of DBN (upper panel) and with strong DBN signals following scratch injury (lower panel). Representative image of three independent experiments. Scale bars: 10 μm. **B** DBN expression analyses by western blotting in enriched astrocyte cultures without, or 1, 4, 16, and 24 h after scratch injury. Quantification of protein levels showing mean, individual data points and SEM; $n = 3$ independent experiments, (one-way ANOVA $F = 23.39$; $DFn = 4$, $DFd = 10$, Dunnett's multiple comparisons test ***$P < 0.001$; multiplicity adjusted $p$ values: no injury vs 16 h: $P = 0.0003$; no injury vs 24 h: $P = 0.0001$). **C** Sections of P30 mouse brains without (upper panels) or with stab injury (lower panels) were labeled with anti-DBN (green) and anti-S100β (upper panel, magenta) or anti-GFAP (bottom panel, magenta). Arrows show astrocytes devoid of DBN (upper panel) and with DBN protein following stab injury (lower panel). Scale bars: 10 μm. Magnifications on the right show no or high expression of DBN in S100ß+ or GFAP+ astrocytes in brains with or without stab injury, respectively. Representative image of two animals, three brain slices/animal. Scale bars: 20 μm. All confocal images in this figure are single optical sections. Source data are provided as a Source Data file.

Cresyl violet stainings supported this idea: multiple non-neuronal (Nissl body-) cells were present in proximity to the injury sites in $Dbn^{-/-}$ brains (Figure S4A). Labeling of stab-wounded brains with an antibody recognizing ALDH1L1, which is expressed in quiescent and reactive astrocytes[25], identified that GFAP-/cytoplasmic SOX9+ cells in $Dbn^{-/-}$ brains express this astrocyte marker protein (Figure S3B). Thus, 30 days after wounding, $Dbn^{-/-}$ astrocytes still occupied lesion sites, but were unable to uphold their reactive astrogliosis program.

Astrocyte scar formation and proliferation after injury are negatively regulated by invading monocytes[4]. We therefore studied the content of CD45+ monocytes in injured WT and $Dbn^{-/-}$ brains by immunohistochemistry (IHC). Comparable numbers of CD45+ monocytes were present inside lesion sites of

WT and $Dbn^{-/-}$ brains (30 DPI), but not in the surrounding tissue (Figure S4B).

At the same time, we identified a substantial loss of NeuN+ and Nissl+ cell bodies surrounding lesions in $Dbn^{-/-}$, but not in WT brains (Fig. 2E, Figure S4A). These findings show long-term neurodegeneration upon cortical stab-injuries in DBN-deficient mice.

Finally, to exclude the possibility that neurodegeneration in $Dbn^{-/-}$ brains following stab injury is owing to generally increased neuronal vulnerability rather than the observed defective astrocyte scarring, we analyzed the outcome of stab injury in the brain of $Dbn^{fl/fl}$:CAMK-Cre$^{+/cre}$ ($Dbn^{fl/fl}$:CAMK-Cre) mice. These mice lose DBN expression in neurons, but not in astrocytes. Following cortical stab injury, $Dbn^{fl/fl}$:CAMK-Cre

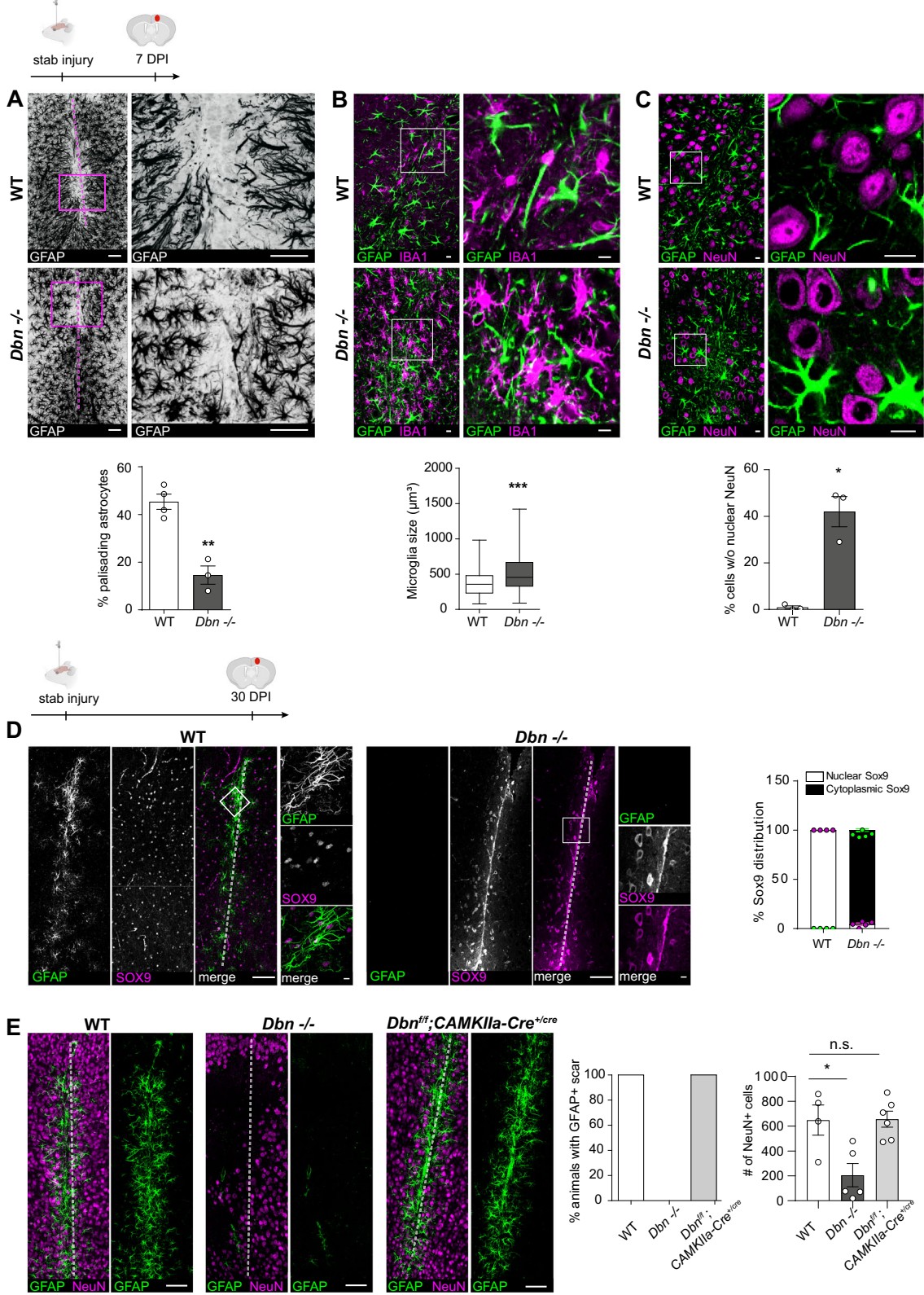

animals consistently show typical GFAP+ astrocytes with SOX9+ nuclei in lesions, with no signs of NeuN translocation after seven days of injury or NeuN loss 30 days after stab injury (Figure S3A, S4C, Fig. 2E). In summary, DBN is essential for astrocyte reactivity and damage containment during traumatic brain injury. DBN-loss perturbs injury-induced scar formation and suppresses maintenance of astrocyte reactivity, providing a causal link of DBN function during injury-induced reactive astrogliosis.

**DBN controls an injury-induced RAB8 membrane compartment.** Towards identifying the DBN-dependent cellular mechanism in the injury setting, we employed the in vitro scratch injury of

**Fig. 2 DBN controls astrocyte scar formation and astrocyte reactivity in vivo. A** GFAP+ astrocytes at core lesion sites in WT (upper panels) and $Dbn^{-/-}$ (lower panels) brains, 7 days post stab injury (DPI). Magenta lines indicate the stab injury from needle insertion. Magnifications show stab lesions, with "palisading" astrocytes extending towards injury sites in WT mice. Scale bars: 100 µm. Quantification of palisading astrocytes in stab wounds 7 DPI, bar graph shows mean, individual data points and SEM. $n = 4$ WT and 3 $Dbn^{-/-}$ animals, Student's unpaired $t$ test, two-sided, $**P = 0.0016$ $t = 6.205$ $df = 5$). **B** IBA1 + microglia (magenta) and GFAP+ reactive astrocytes (green) in the periphery of scars 7 DPI, in WT (upper panel) and $Dbn^{-/-}$ brains (lower panel). Scale bars: 10 µm. Quantification of soma sizes of IBA1+ microglia. $n = 145$ WT & 122 $Dbn^{-/-}$ cells from three mice per condition, graph shows box and whisker plots: box extends from 25th to 75th percentiles, central line = median, whiskers comprise all values from minimum to maximum, $F$ test $*** < 0.0001$, Unpaired $t$ test with Welch's correction, two-sided; $***P = 0.000000205637175$, $t = 5.381$ $df = 201$). **C** NeuN+ neurons (magenta) and GFAP+ reactive astrocytes (green) in scars of WT (upper panel) and $Dbn^{-/-}$ brains (lower panel) 7 DPI. Scale bars: 10 µm. Quantification shows neurons without nuclear NeuN in scars. $n = 3$ WT and three $Dbn^{-/-}$ mice, bar graph shows mean, individual data points and SEM, Unpaired $t$ test with Welch's correction, two-sided; $*P = 0.0231$, $t = 6291$ $df = 2044$). **D** IHC of WT and DBN$^{-/-}$ brains with 30-day post stab lesions. GFAP labeling (green) detects reactive astrocytes; SOX9 labeling (magenta) detects astrocytes. Scale bars in overview images: 200 µm. Scale bars in close up images: 10 µm. Quantification of cytoplasmic/nuclear SOX9 distribution in astrocytes in lesion sites. Single data points of nuclear SOX9 and corresponding error bars are displayed in magenta, cytoplasmic SOX9, and corresponding error bars in green. $n = 4-5$ animals per condition. **E** NeuN+ neurons (magenta) and GFAP+ reactive astrocytes (green) in WT (left), $Dbn^{-/-}$ (center) and $Dbn^{fl/fl}$:CAMK-Cre$^{+/cre}$ brains (right), 30 DPI. Quantifications show the percentage of animals with GFAP+ astrocytes (left graph) and NeuN+ cells (right graph) at lesion sites 30 days post stab wounding. Bar graph shows mean, individual data points and SEM $n = 4$ WT, five $Dbn^{-/-}$ animals, six $Dbn^{fl/fl}$:CAMK-Cre animals, (one-way ANOVA D$Fn = 2$, $DFd = 12$, $F = 8.397$; Dunnett's multiple comparisons test Multiplicity adjusted $p$ value: $*P = 0.0112$). All images in this figure are confocal stacks. Scale bars: 200 µm. Source data are provided as a Source Data file.

cultured astrocytes. Following injury, WT astrocytes polarized, and extended long processes into the injury site (Figure S5A). In contrast, the majority of $Dbn^{-/-}$ astrocytes extended over smaller distances and frequently changed their orientation (Figure S5A; Movie 2), resulting in a significant decrease in wound closure (Figure S5B). During the wounding response, DBN, as well as overexpressed DBN-YFP only partially localized to prominent actin fibers and also decorated internal compartments consisting of vesicular and tubular structures (Figures S5C, D).

Given this subcellular distribution in astrocytes, together with a previous observation that identified DBN in association with internal membrane structures[26], we asked if DBN mediates injury responses by regulating membrane trafficking. Analyses of several trafficking compartments, including RAB5+, RAB7+, or RAB11 + endosomes showed no major differences in $Dbn^{-/-}$ astrocytes when compared with WT cells (Figure S6A). However, DBN-loss severely disturbed the distribution of RAB8 in astrocytes during mechanical injury. In WT astrocytes, using either GFP-RAB8A or an antibody that detects both RAB8A and RAB8B, we obtained signals of prominent RAB8+ tubular compartments adjacent to injury sites (Fig. 3A, Figure S6A). No signals were detected in cells depleted for both RAB8A and RAB8B demonstrating specificity to the antibody (Figures S46B, C). In $Dbn^{-/-}$ astrocytes, instead of marking internal tubules, RAB8 was mostly dispersed throughout the cytosol (Fig. 3A).

Western blotting (WB) confirmed that RAB8 protein levels and GTPase activity in $Dbn^{-/-}$ astrocytes were not altered when compared with WT, irrespective of injury (Fig. 3B, C), indicating that DBN is required during the generation of the tubule compartment rather than influencing the activity of the GTPase. Analogous to endogenous RAB8, GFP-RAB8A highlighted prominent membrane tubules in WT astrocytes, whereas GFP-RAB8A tubules in $Dbn^{-/-}$ were fragmented (Fig. 3D, Movie 3). Quantification of tubules in uninjured astrocyte cultures revealed very few RAB8A-positive tubules in both WT and $Dbn^{-/-}$ astrocytes. However, mechanical scratch injury increased the tubule number by 2.5-fold in WT astrocytes but not in $Dbn^{-/-}$ astrocytes (Fig. 3E), which indicates that tubular RAB8A endosomes form in response to injury and depend on the presence of DBN. We also detected Rab8+ tubules in polarizing WT astrocytes after injury in 21 DIV mixed cortical cultures and 7 DPI in vivo (Fig. 3F, G). In contrast, in $Dbn^{-/-}$ astrocytes, RAB8 was dispersed in processes and largely accumulated in vesicular

structures in the cell bodies. Only occasionally, RAB8 associated with short tubular structures. These findings suggest that DBN-dependent RAB+ tubular structures are regulators of astrogliosis in the setting of brain injury and scarring responses in vivo.

**DBN generates tubular endosomes by counteracting Arp2/3-dependent actin nucleation.** Dynamic rearrangement of the actin cytoskeleton is a critical component in generating tubular membrane systems[27,28]. It relies on the capacity of actin filaments to polymerize and depolymerize, a characteristic that can be influenced by the ability of DBN to bind the lateral filament surface, to bundle actin filaments and to inhibit actin nucleation[6,29,30]. We analyzed whether defects of $Dbn^{-/-}$ astrocytes in RAB8 compartments are linked to DBN and its function as actin regulator. In the first set of experiments, we co-transfected astrocyte cultures isolated from $Dbn^{-/-}$ mice with DBN-YFP (or control YFP) and labeled RAB8A using red fluorescent mRuby-RAB8A. Following in vitro injury, DBN-YFP expression, but not YFP, rescued the ability of $Dbn^{-/-}$ astrocytes to form membrane tubules (Fig. 4A). Next, we employed pharmacological approaches to address whether defective actin dynamics disrupts Rab8a tubules in $Dbn^{-/-}$ astrocytes. Low concentrations of Cytochalasin D (100 nM) interfere with the nucleation of new actin filaments whilst leaving existing filaments unaffected[31]. During scratch injury in $Dbn^{-/-}$ reactive astrocytes, this Cytochalasin D concentration re-established RAB8A-positive membrane tubules (Fig. 4B, Movie 4), suggesting that DBN induces tubule formation by antagonizing the cellular actin nucleation machinery. In this model, loss of DBN would increase actin nucleation. To test this idea, we applied the ARP2/3 inhibitor CK-666 or the formin-inhibiting molecule SMIFH2[32,33] to GFP-RAB8A expressing $Dbn^{-/-}$ astrocytes. Although inhibition of the ARP2/3 nucleation machinery by CK-666 reinstalled the ability of $Dbn^{-/-}$ astrocytes to form RAB8A tubules, the formin inhibitor did not (Fig. 4B Movie 4). Thus, DBN stabilizes actin filaments by mechanisms involving suppression of ARP2/3-dependent actin nucleation, a process essential for the injury-induced formation of the tubular RAB8 compartment.

**DBN loss affects uptake and intracellular distribution of plasma membranes.** RAB8 compartments have previously been reported to engage in plasma membrane recycling as well as in Golgi-derived delivery of cargo to the plasma membrane[34].

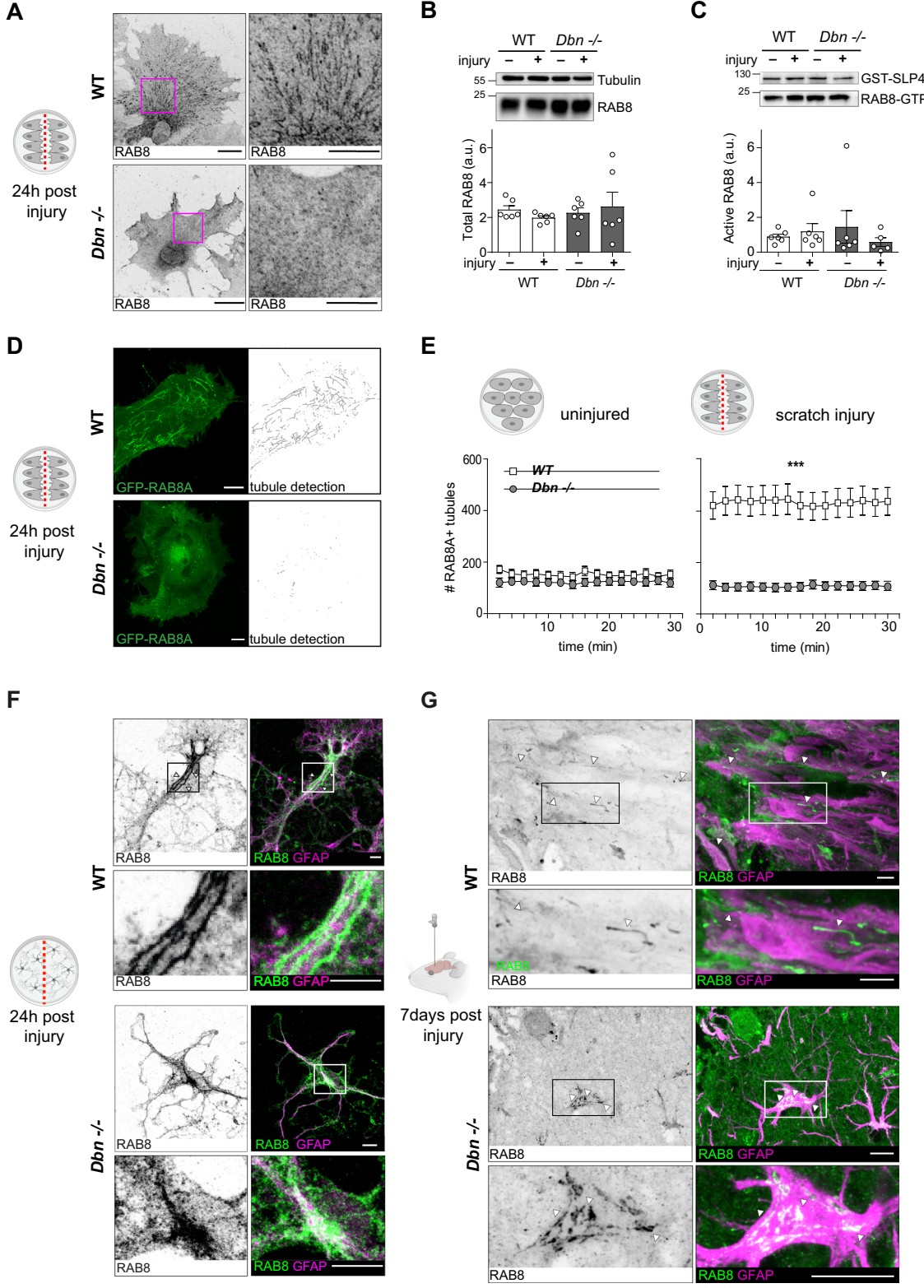

To characterize the specificity of endosomal trafficking of the DBN-dependent RAB8 compartment, we analyzed endogenous RAB8 tubules in WT and $Dbn^{-/-}$ astrocytes after injury using immunocytochemistry. RAB8-positive tubular structures were clearly orientated and extended through the cytosol of WT astrocytes, aligning closely with prominent actin filaments (Fig. 4C). In contrast, anti-RAB8 labeling in $Dbn^{-/-}$ astrocytes produced a diffuse cytosolic signal, with the appearance of cistern-like structures in the cellular periphery. These structures possessed a prominent ring of filamentous actin adjacent to the RAB8 signal, from where stub-like tubules originated (Fig. 4C). The presence of peripheral membrane cisterns suggested a general ability of $Dbn^{-/-}$ astrocytes to endocytose. We provoked increased membrane uptake in GFP-RAB8A-expressing astrocytes via EGF-induced macropinocytosis[35].

**Fig. 3 DBN controls the formation RAB8 tubular endosomes during injury in cultured astrocytes. A** RAB8 labeling in WT (upper panels) and $Dbn^{-/-}$ astrocytes (lower panels) 24 h after scratch injury in vitro. Scale bar: 10 μm. Representative images of four independent experiments. Images are confocal stacks. **B** Western blot analysis of RAB8 protein in WT and $Dbn^{-/-}$ astrocytes without or 24 h after injury. Tubulin serves as loading control. Rab8 and Tubulin antibody incubations were performed on the same membrane on consecutive days without stripping. Graph shows quantification of RAB8 protein levels in WT and $Dbn^{-/-}$ astrocytes, displaying mean, individual data points, and SEM ($n = 6$ independent experiments, Bartlett's test for equal variances *** $P = 0.0004$, one-sided Kruskal–Wallis test with Dunn's multiple comparisons test $P = 0.6048$. **C** Western blot analysis of active RAB8-GTP isolated by GST-SLP4 pulldown from lysates obtained from WT and $Dbn^{-/-}$ astrocytes without or 24 h after scratch injury. GST levels confirm equal amounts of SLP4 in pulldowns. Rab8 and GST antibody incubations were performed on the same membrane on consecutive days without stripping. Quantification shows GTP-bound RAB8 levels in WT and $Dbn^{-/-}$ astrocytes, displaying mean, individual data points and SEM ($n = 6$ independent experiments, Bartlett's test for equal variances **$P = 0.0014$; one-sided Kruskal–Wallis test with Dunn's multiple comparisons test $P = 0.3589$). **D** GFP-RAB8A distribution in transfected WT (upper panels) and $Dbn^{-/-}$ astrocytes (lower panels). GFP-RAB8A-labeled compartments were detected automatically and are shown as skeletonized structures (black). Representative images of three independent experiments. Scale bars: 10 μm. Images are confocal stacks. **E** Quantification of GFP-RAB8A+ tubules in WT and $Dbn^{-/-}$ astrocytes, uninjured ($n = 9$–10) and 24 h after scratch injury, displaying mean and SEM ($n = 28$–32, two-way ANOVA (repeated measurements), F = 35.7, $DFn = 1$, $DFd = 58$, Bonferroni's multiple comparisons test ***$P < 0.001$ for all timepoints; multiplicity adjusted p values for timepoints in consecutive order: 0.000000341871; 0.000000021531; 0.000000016513; 0.000000039208; 0.000000017378; 0.000000014879; 0.000000010948; 0.000000189338; 0.000000625271; 0.000000265748; 0.000000100183; 0.000000085847; 0.000000034144; 0.000000095626; 0.000000034095). Cells were imaged for 30 min. **f** Presence of Rab8+ tubules (green) in GFAP+ astrocyte (magenta) at the wound edge of injured 21 DIV cortical cultures (upper panel). RAB8 in corresponding $Dbn^{-/-}$ astrocytes was dispersed in processes but accumulated in cell bodies (bottom panel). Representative images of three independent experiments. Scale bars: 10 μm. Confocal images are single optical sections. **G** Presence of Rab8+ tubules (green) in palisading processes of GFAP+ WT astrocytes (magenta) at lesion core following stab injury in the brain (upper panel). Arrowheads show Rab8+ tubules. RAB8 accumulated in distinct structures in soma of $Dbn^{-/-}$ astrocytes (arrows), whereas their processes showed no tubules (bottom panel). The tissue was labeled at 7 DPI. Representative images of two animals per condition. Scale bars: 10 μM. Images are confocal stacks. Source data are provided as a Source Data file.

EGF-treated WT astrocytes presented throughout the cytosol prominent GFP-RAB8A+ tubules, which locally assembled via the gathering of small Rab8A-positive particles (Movies 5 and 6). In $Dbn^{-/-}$ astrocytes, RAB8A+ membranes accumulated beneath the plasma membrane, where membrane cisterns frequently emerged but also rapidly disappeared (Figures S7A, Movies 5 and 6). These results establish the function of DBN during the generation and distribution of RAB8 associated membrane tubules.

Given that stabilization of membrane tubules requires the microtubule cytoskeleton in other cell types[36–38], we considered whether excessive ARP2/3-dependent actin dynamics in $Dbn^{-/-}$ astrocytes (Fig. 4B) antagonize the association of microtubules with the RAB8 compartment. Here, we visualized microtubules using SiR-tubulin in GFP-RAB8A expressing $Dbn^{-/-}$ astrocytes (Fig. 4D, Movie 7). GFP-RAB8A vesicles were visible in the cell's periphery in proximity to microtubules, but they did not form tubules. However, during the administration of CK-666, RAB8A-positive vesicles turned immediately into tubules, which then extended along the microtubules towards the cell body (Fig. 4D). Thus, DBN controls RAB8 membrane trafficking by antagonizing ARP2/3-dependent actin networks to enable the transport of tubular membranes along microtubules. In turn, DBN-loss disturbs the normal actin equilibrium and creates excessive ARP2/3 activity, which prevents the microtubule-assisted formation of tubular endosomes from RAB8+ vesicles (Fig. 4E).

**DBN-dependent RAB8A tubules are required for β1-integrin trafficking.** A known RAB8 target in cell lines is β1-integrin, a key molecule in cell polarization and migration[39,40], which also controls astrocyte differentiation and reactivity[41–43]. We used a C-terminus-specific β1-integrin antiserum to detect the spatio-temporal distribution of β1-integrin in astrocytes during injury, and identified prominent β1-integrin foci in RAB8 membrane tubules (Fig. 5A). To examine if DBN regulates β1-integrin, as a first read-out, we exploited the composition and assembly of focal adhesions, which are hot spots of β1-integrin activity. Antibody labeling in WT astrocytes during scratch injury showed the concentration of active β1-integrin, as well as the intracellular adapter paxillin in mature focal adhesions[40]. $Dbn^{-/-}$ astrocytes, in contrast, exhibited scattered membrane distribution of active β1-integrin with smaller paxillin+ focal adhesions (Fig. 5B). By

tracking GFP-tagged paxillin, we identified a persisting reduction in focal adhesion sizes during the injury-induced polarization of $Dbn^{-/-}$ astrocytes, when compared to WT cells (Fig. 5C). Thus, DBN controls the presentation of β1-integrin to focal adhesions in astrocytes during their responses to injury.

We then investigated if the DBN-dependent, injury-induced RAB8 membrane tubules are required for β1-integrin trafficking. To study the distribution and trafficking of β1-integrin independent of conformational states and antibody epitopes, we performed surface biotinylation experiments in injured WT and $Dbn^{-/-}$ astrocyte cultures. Although signals of intracellular biotinylated β1-integrin were at the detection limit after internalization, we found a substantial reduction of surface β1-integrin in $Dbn^{-/-}$ astrocytes when compared with WT astrocytes (Fig. 5D). Finally, we followed the internalization of active integrin in situ with an antibody-feeding assay. Thirty minutes after antibody labeling with anti-β1-integrin, the receptor was widely distributed throughout WT astrocytes (Fig. 5E). In contrast, in $Dbn^{-/-}$ astrocytes, β1-integrin accumulated beneath the plasma membrane of polarizing cells. We obtained a comparable β1-integrin distribution in WT astrocytes after depleting both RAB8 isoforms with RNAi (Fig. 5E), suggesting that DBN-dependent RAB8 compartments are major routes for β1-integrin trafficking. We conclude that the DBN-dependent, injury-induced RAB8 tubule compartment functions as an important hub to distribute internalized β1-integrin in astrocytes during injury.

**DBN loss induces intracellular membrane accumulation in reactive astrocytes in vivo.** The extensive accumulation of RAB8+ membranes beneath the leading edge of cultured $Dbn^{-/-}$ astrocytes prompted us to analyze internal membranes in vivo at the ultrastructural level using high-resolution transmission electron microscopy (TEM). We compared WT and $Dbn^{-/-}$ brains from 7 days post stab injury, using GFAP+ to locate reactive astrocytes in lesions. Processes of reactive astrocytes in WT brains in proximity to the stab lesion contained several endosome-like organelles, which frequently connected to thin membrane tubules. In contrast, processes of $Dbn^{-/-}$ reactive astrocytes in proximity to the stab injury displayed massive cytoplasmic vacuolization. The vacuoles resembled multilamellar bodies, as

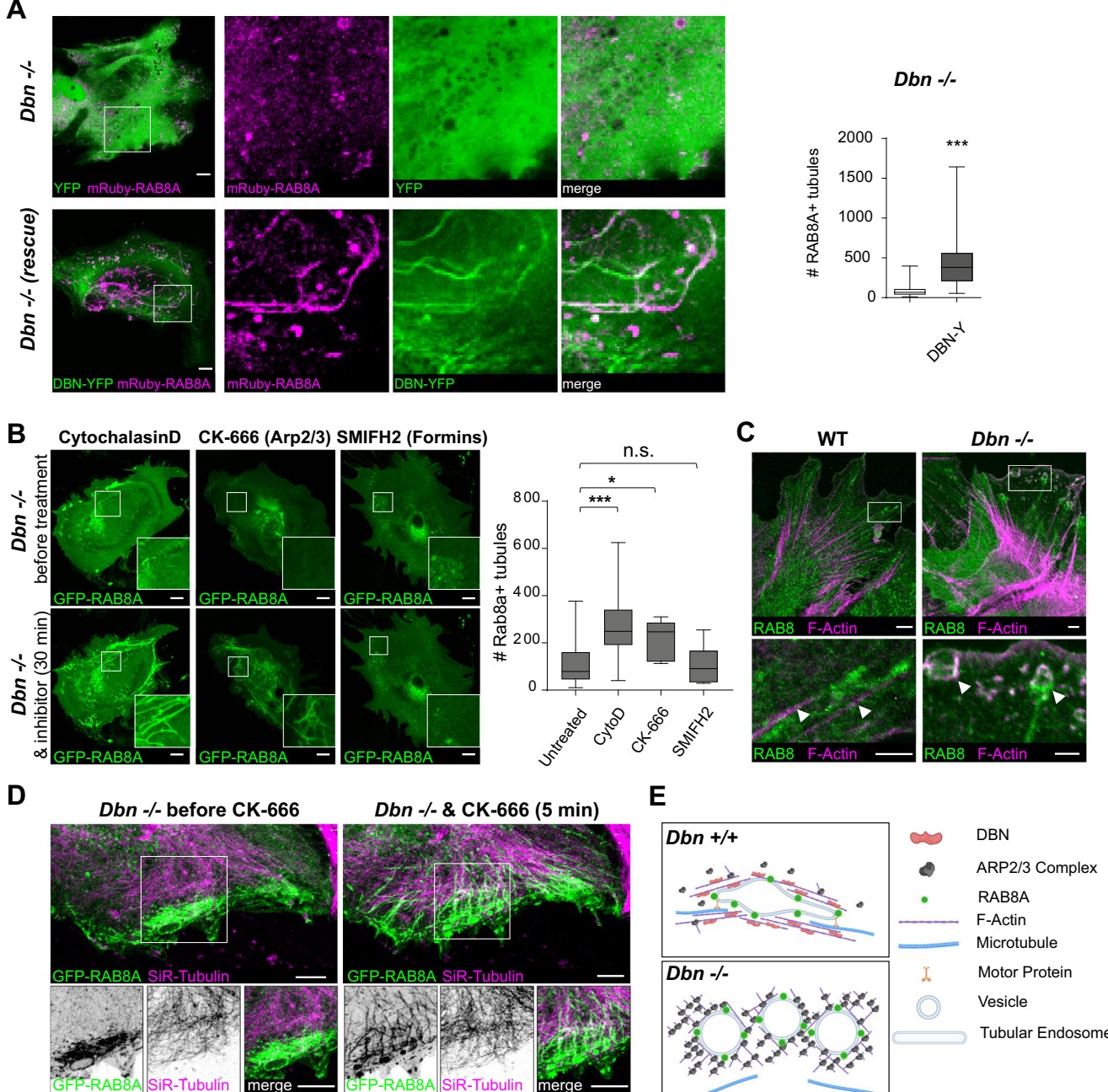

**Fig. 4 DBN balances the actin nucleation machinery during RAB8A tubule formation. A** DBN rescues RAB8A membrane tubules in $Dbn^{-/-}$ astrocytes. $Dbn^{-/-}$ astrocytes co-transfected with mRuby-RAB8A and YFP-control (upper panels), or with mRuby-RAB8A and DBN-YFP (bottom panels). Scale bars: 10 μm. Images are confocal stacks. Quantification of mRuby-RAB8A tubules displayed as box and whisker plots: box extends from 25th to 75th percentiles, central line=median, whiskers comprise all values from minimum to maximum; $n = 63$–65 cells from three independent experiments ($F$ test for equal variances ***$P < 0.0001$, Unpaired $t$ test with Welch's correction, two-sided, ***$P = 0.000000000025$, $t = 7.965$ $df = 68.19$). **B** Live imaging of $Dbn^{-/-}$ astrocytes expressing GFP-RAB8A before (upper panels) and 30 min after treatment with different inhibitors (lower panel); 100 nm Cytochalasin D (left), 100 μm CK-666 (center), or 25 μm SMIFH2 (right). Scale bars: 10 μm. Images are confocal stacks. Graph shows the quantification of GFP-RAB8A tubules 30 min after inhibitor treatment, Cytochalasin D, $n = 16$; SMIFH2, $n = 8$; CK-666, $n = 9$ cells; all from three experiments. Box and whisker plots: box extends from 25th to 75th percentiles, central line=median, whiskers comprise all values from minimum to maximum (one-way ANOVA $F = 11.47$, $DFn = 3$, $DFd = 63$, Dunnett's multiple comparisons test, multiplicity adjusted $p$ values: * $P = 0.01534048$, ***$P = 0.0001$, ns $P = 0.9999$). **C** Distribution of endogenous RAB8A (green) and F-Actin (magenta) in WT (left panel) and $Dbn^{-/-}$ (right panel) astrocytes. Arrowheads in WT panel indicate RAB8+ tubules aligned with actin fibers. In the $Dbn^{-/-}$ panel, arrowheads highlight RAB8+ membrane cisterns. Representative images from three different experiments. Scale bars: 10 μm. Images are single confocal sections. **D** $Dbn^{-/-}$ astrocytes expressing GFP-RAB8A and labeled with SiR-tubulin before (left panel) and 5 min after adding 100 μM CK-666 (right panel); see Movie 7. Scale bars: 10 μm. RAB8+ tubules formed immediately after adding CK-666 and aligned with adjacent microtubules, while extending into the cytosol. Representative images showing the effect observed by CK-666 treatment in three independent experiments. Images are single confocal sections. **E** Proposed mechanism.: DBN functions as an essential switch in the actin network homeostasis, which supports the formation of RAB8A + tubular endosome along microtubules. Source data are provided as a Source Data file.

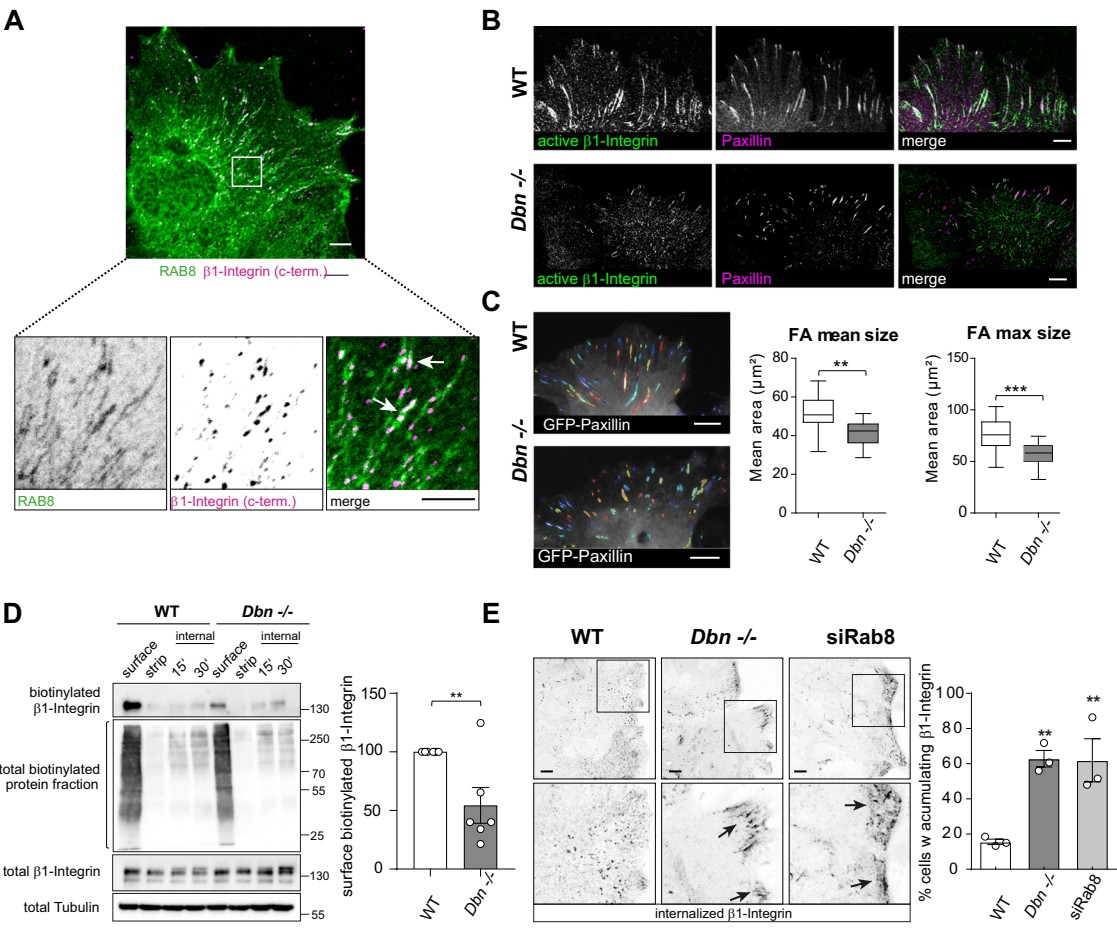

**Fig. 5 DBN regulates RAB8 membrane trafficking of β1-integrin during injury. A** Detection of β1-integrin (magenta in merged images) in RAB8+ tubules (green in merged images) in astrocytes. Arrows indicate RAB8+ tubules enriched with β1-integrin. Representative image of three independent experiments. Scale bar: 10 μm. Confocal images are single optical sections. **B** Confocal images of WT (upper panel) and *Dbn*$^{-/-}$ astrocytes (bottom panel) after mechanical injury, labeled for active β1-integrin (green) and paxillin (magenta). Representative images of three different experiments. Scale bars: 10 μm. Images are single optical sections. **C** Focal adhesions (FA) in WT (upper panel) and *Dbn*$^{-/-}$ astrocytes (lower panel) expressing GFP-paxillin analyzed by the Focal Adhesion Analysis Server. Graph shows box and whisker plots (box extends from 25th to 75th percentiles, central line=median, whiskers comprise all values from minimum to maximum) of focal adhesion (FA) mean sizes and FA maximum sizes over 22 h during live imaging experiments (*n* = 17 cells from three independent experiments, Student's unpaired *t* test (two-sided): FA mean size **P = 0.0013, t = 3.515 df = 32; FA max size ***P = 0.0004, t = 3.995 df = 32). Scale bars:10 μm. **D** Streptavidin-pulldown of proteins after surface biotinlylation of injured WT and *Dbn*$^{-/-}$ astrocytes. Western blot shows biotinylated β1-integrin in different conditions: immediately after surface biotinylation (surface), after removal of the surface-biotin label with MESNA (strip), 15 min and 30 min after incubating labeled astrocytes at 37 °C followed by MESNA treatment to remove the fraction of surface-exposed labeled proteins. Bar diagram shows quantification of surface β1-integrin after normalization to total integrin levels displaying mean, individual points and SEM; *n* = 6 experiments, Student's unpaired *t* test (two-sided) **P = 0.0054, t = 3.541, df = 10. **E** Labeling of internalized ligand-bound β1-integrin following antibody feeding in WT astrocytes (left image), *Dbn*$^{-/-}$ astrocytes (center image), and WT astrocytes after siRNA depletion of RAB8A and RAB8b (right image). Arrows indicate accumulations of β1-integrin directly beneath the leading edge of *Dbn*$^{-/-}$ or RAB8-depleted astrocytes. Boxes in upper panels are magnified in lower panels. Images are confocal stacks. Bar diagram shows quantification of astrocytes with antibody-labeled β1-integrin at leading edges displaying mean, individual points and SEM;. *n* = 20–37 cells from three experiments; one-way ANOVA F = 12.57, DFn = 2, DFd = 6 with Dunnett's multiple comparisons test WT vs *Dbn*$^{-/-}$: **P = 0.0083, WT vs siRab8: **P = 0.0091. Source data are provided as a Source Data file.

they were filled with amorphous, electron-dense material that was surrounded by double or multiple concentric membrane layers (Fig. 6A)[44]. Quantification of multilamellar bodies at injury and contralateral sites of WT and *Dbn*$^{-/-}$ brains showed that this compartment was induced upon injury only in the absence of DBN (Fig. 6A, B). In addition, we investigated the ultrastructure of astrocytic endfeet, which sustain high levels of membrane trafficking during nutrient and metabolite uptake, and efflux of waste products[2]. Analogous to astrocyte processes, astrocyte endfeet at the injury site in WT brains contained several endosomal structures. In contrast, excessive amounts of multilamellar bodies filled the endfeet at injury sites in *Dbn*$^{-/-}$ mice

(Figure S7B). In summary, astrocytes from *Dbn*$^{-/-}$ mice show excessive, injury-specific accumulation of membrane-derived material in both astrocytic processes and endfeet.

## Discussion

We have identified the function of DBN in protecting the brain from tissue damage following injury. We show that (1) under physiological conditions, DBN protein is not expressed in astrocytes; however, its injury-induced upregulation in reactive astrocytes is required for the coordinated formation and maintenance of astrocyte scars, demonstrating its essential role in effective tissue protection in the brain. (2) At the cellular level, DBN shifts the

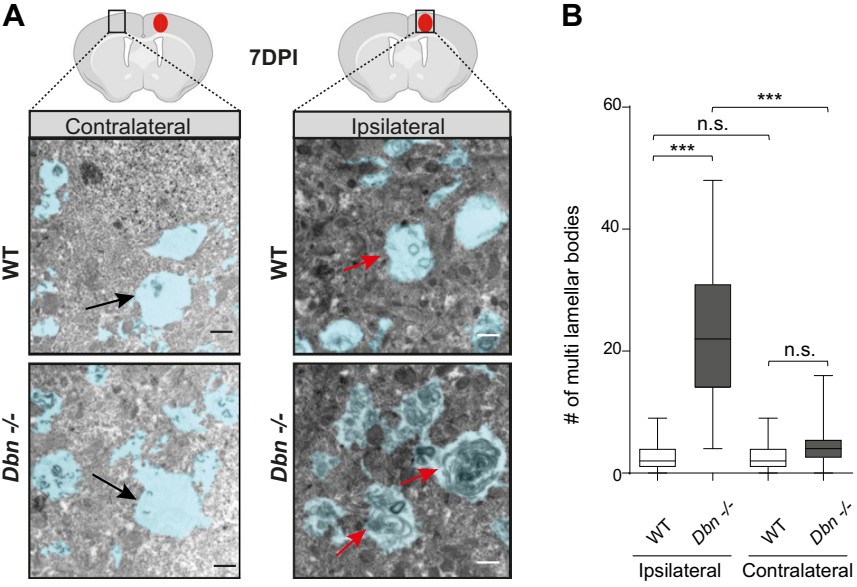

**Fig. 6 DBN loss induced membrane accumulations in polarizing astrocytes during injury. A** TEM ultrastructural architecture of astrocytes in brains of WT and $Dbn^{-/-}$ mice. Images were acquired on the contralateral side of injury (left panels) or inside the lesion core (right panels); 7 days post stab injury (7 DPI). Astrocyte processes are shaded in blue. Black arrows show astrocyte processes largely devoid of membranous compartments, on the site contralateral to the stab injury. Red arrows show astrocyte processes at the injury site with endosome-like organelles (WT, upper image) or with multilamellar bodies ($Dbn^{-/-}$, bottom image). Scale bars: 500 nm. **B** Quantification of multilamellar bodies displayed as box and whisker plots: box extends from 25th to 75th percentiles, central line=median, whiskers comprise all values from minimum to maximum. Field of view size: 1170 µm². Number of fields/view: 92 WT and 77 $Dbn^{-/-}$ ipsilateral, 85 WT and 77 $Dbn^{-/-}$ contralateral from three WT and three $Dbn^{-/-}$ animals (Bartlett's test for equal variances ***$P < 0.001$; one-sided Kruskal–Wallis test with Dunn's multiple comparisons test ***$P < 0.001$; multiplicity adjusted $p$ values: Ipsilateral WT vs Ipsilateral $Dbn^{-/-}$: $P = 0.000000000000047$; ipsilateral WT vs contralateral WT: $P = 0.999999999999999$; contralateral WT vs contralateral $Dbn^{-/-}$: $P = 0.190172692939112$; Ipsilateral $Dbn^{-/-}$ vs contralateral $Dbn^{-/-}$: $P = 0.000000755492543$). Source data are provided as a Source Data file.

actin network organization from ARP2/3-dependent arrays to microtubule-compatible scaffolds, which facilitate the formation of injury-induced RAB8A-positive membrane tubules. (3) These tubules serve as a hub for the membrane trafficking of surface proteins involved in coordinating adhesive responses, such as β1-integrin. (4) The actin-dependent facilitation of RAB8 membrane tubules by DBN is essential to maintain astrocyte reactivity, ensure the physical integrity of the scar and prevent neurodegeneration.

We, therefore, propose a conceptually new role for DBN as an injury-induced actin regulator in reactive astrocytes.

**DBN functions as an injury-induced actin regulator in membrane trafficking.** DBN is highly abundant in dendritic spines and developing neurites, and confers resilience in dendritic spines during cellular stress responses[13,45,46]. Surprisingly, DBN protein is not detectable in astrocytes under healthy conditions but immediately upregulated after mechanical damage (Fig. 1, Figure S1). These findings are corroborated by previously published transcriptome data: focal penetrating injuries, like stab wounds and spinal cord injuries, cause significant upregulation in DBN transcripts in astrocytes. However, other disease models without direct mechanical damage, like "transient middle cerebral artery occlusion" -based stroke or lipopolysaccharide injections, show less pronounced increases in DBN mRNA[47–49]. Tissue damage in association with lesions is thus one major trigger for DBN upregulation in astrocytes. Currently, neither the molecular signaling pathways nor the mechanisms controlling DBN protein upregulation, are known. However, it is conceivable that DBN protein abundance is, in addition to increased transcription, also modulated by post-translational modifications. Such mechanisms would, analogous to the ATM-dependent phosphorylation of DBN in neurons[13], extend DBN protein lifetime and abundance.

Unexpectedly, the cell biological mechanism underlying the role of DBN during astrocyte scar formation involves a directive function for its participation in membrane trafficking. Until now, few studies have linked DBN to membrane dynamics in cells, other than a demonstration that DBN principally binds to membranes[26], that it restricts the entry of rotavirus into cells by limiting endocytosis[50], and that it regulates antigen presentation of dendritic cells[51]. Our study identifies a further avenue in DBN-dependent membrane trafficking and protein sorting mechanism that functions by stabilizing RAB8-positive tubular endosomes upon injury.

In cell culture and in vivo, we discovered prominent RAB8-positive tubular endosomes specific to a post-injury setting in astrocytes, which extend long palisading-like processes into wound areas. The GTPases, RAB8A and RAB8B, facilitate distinct routes of polarized membrane transport. RAB8 is involved in endocytosis, membrane recycling, autophagy, and exocytosis by associating with vesicles, macropinosomes, and tubules[34,52,53]. Moreover, RAB8 has been implicated in various diseases ranging from microvillar inclusion disease and cancer to neurodegenerative diseases such as Alzheimer's and Parkinson's disease[34]. Tubular membrane compartments serve as logistical platforms to sort membrane-bound cargos[54]. Membrane tubulation occurs particularly on maturing early endosomes and macropinosomes to separate components from the quick "bulk flow" back to the plasma membrane and to direct them towards other compartments such as the Golgi network or recycling endosomes[55]. The actin cytoskeleton creates stable tubular subdomains, where membrane-bound cargos segregate and concentrate according to their designated destinations[28,56,57]. We propose a role for DBN in inward-directed RAB8-based membrane trafficking: upon wounding, DBN may contribute to the uptake of plasma membrane material by compiling

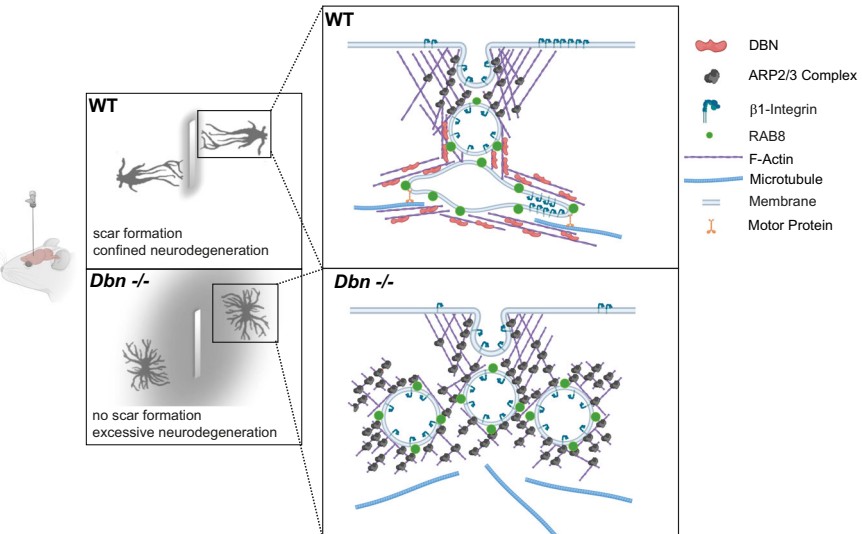

**Fig. 7 Astrocytes require DBN to control damage during brain injury. Proposed model of how DBN/Rab8 functions during astrocyte polarization and scarring responses.** In response to CNS injury, DBN generates actin scaffolds suitable for tubule-based membrane trafficking. DBN-loss results in the accumulation of endocytotic vesicles beneath the plasma membrane and affects the sorting of adhesion receptors essential for interaction with lesion core components. In this way, DBN/Rab8 membrane tubules are fundamental for the polarization and scarring responses of reactive astrocytes.

endocytosed receptors and material in the injury-induced RAB8 compartment that are essentially sorted by DBN-stabilized tubules (Fig. 7).

**DBN antagonizes ARP2/3-dependent actin dynamics.** ARP2/3-dependent actin nucleation has been shown to drive the fission of existing tubules into vesicular endosomes by the WASH complex[58]. DBN associates dynamically with scaffolds of forming tubular endosomes and counteracts their ARP2/3-dependent fission (Fig. 4b, d). DBN could thereby occlude ARP2/3-binding sites on the actin scaffold by its sidewise binding to actin filaments[6]. This model is consistent with DBN competition of ARP2/3 function, as identified in our pharmacological rescue experiments. The colocalization of DBN along RAB8+ tubules in our rescue experiments further supports a role of DBN as an integral scaffold component around tubular endosomes. Structural analyses showed that drebrin is able to form tetramers and thereby to bundle actin filaments[30]. An analogous organization of actin filaments by DBN at RAB8+ compartments could possibly promote membrane tubulation as well as counteract the ARP2/3 machinery. In addition, DBN could also facilitate the extension of RAB8 tubules from underneath the leading edge into the cell body by arranging the surrounding actin architecture. The observable actin filaments in WT astrocyte processes generally run parallel to RAB8 tubules. These linear actin filaments may serve as a permissive scaffold for RAB8 tubules, which then stabilize along intermingled microtubules analogous to membrane tubulation during sorting to recycling endosome transition in non-neuronal cells[36–38]. This mechanism would be susceptible to mislocalized ARP2/3-actin arrays upon DBN loss. Our live-imaging results also support this mechanism in view of $Dbn^{-/-}$ astrocytes, where, until their pharmacological disruption, ARP2/3-actin networks prevent the tubulation and transport of RAB8A-positive membrane along microtubules. We propose that DBN serves as a master switch to enable the assembly of microtubule-compatible actin networks for tubular membrane trafficking (Fig. 7).

**Cellular trafficking by RAB8 tubular endosomes.** Tubular endosomes and/or tubule-derived vesicles deliver their content to

intracellular destinations, where it may be redistributed or degraded according to cellular requirements for efficient polarized astrocyte outgrowth. DBN deficiency disrupts the RAB8-dependent tubular protein sorting machinery. Consequently, membranes and associated proteins accumulate in intermediate endosomal compartments instead of being rerouted in a polarized manner in sufficient numbers. The resulting deficits and mis-localization of surface receptors first evoke the failed wounding response and polarization of astrocytes followed by their erratic behavior during scarring. The disarray within the scarring astrocytes could lead to the downregulation of their reactivity, as shown by the loss of GFAP and translocation of SOX9—either in a cell-autonomous manner and/or by an affected interplay with other cell types such as microglia.

Our protein traffic-based mechanism is supported by findings in DBN-depleted epithelial cells, which share some localization defects in apical markers with RAB8A-deficient intestinal cells during microvillar inclusion disease[59,60]. In addition, RAB8 was shown to organize cell adhesion in conjunction with RAB13 by transporting adhesion molecules, in cultured epithelial cells[61]. Moreover, the closely RAB8-related RAB13 organizes the collective cell migration of these non-neuronal cells in wounding assays in an ordered manner[62], analogous to our findings with erratic DBN-deficient astrocytes in scars in vivo and during live imaging experiments.

**β1-integrin trafficking and astrocyte reactivity.** The induction of tubular endosomes in response to injury suggests that they represent an efficient way of transporting and sorting cargo relevant for the injury response in astrocytes. We show the mislocalization of endogenous β1-integrin in DBN-deficient astrocytes in a Rab8-dependent manner. Cells require tight control of β1-integrin in terms of levels and distribution at the cell surface to migrate efficiently. The key mechanism for this is retrograde trafficking[63]. Inadequate β1-integrin subunits at the surface impair the assembly and turnover of focal adhesions, as seen with the outgrowth defects observed in DBN-deficient astrocytes. In particular, the vanishing astrocyte reactivity highlights comprehensive defects in signal reception and computation through the mis-sorting of membrane receptors. Astrocytic

β1-integrin has an important role as a co-receptor in signaling pathways, which control astrocyte reactivity[41–43]. However, other misregulated membrane receptors undoubtedly also contribute to the observed phenotype, and further research screening DBN-deficient astrocytes is required to identify them.

**DBN function in brain astrocytes.** A major finding of this study is the relevance of injury-induced DBN in forming functional astrocyte scars. We showed that DBN is crucial for the coordinated polarization and palisade-like outgrowth of astrocytes. These locally occurring morphological changes are of vital importance to restrict inflammation and fibrotic tissue[64], as astrocytes from adjoining regions do not migrate into lesion sites[15,65]. The DBN-dependent establishment of an astrocyte scar in turn, is essential for damage containment in the CNS, as highlighted by the high neurodegeneration observed in DBN-deficient mice without astrocyte scars. What is more, the severity of the DBN-dependent membrane-trafficking defects became apparent in our in vivo TEM studies. We showed the prominent enrichment of membranous material in multilamellar structures within processes and endfeet of DBN-negative astrocytes. The extent of these accumulating multilamellar structures in astrocytes is, to our knowledge, unique, occurs only upon injury and supports our results of trafficking defects in cell culture. Moreover, this phenomenon resembles membrane whorls, which are evocative for autophagy and which accumulate in smooth muscle cells during atherosclerosis[44,66]. Analogous structures have been reported from Alzheimer's disease and amyotrophic lateral sclerosis patients as well as neurons expressing mutated huntingtin[67–69]. Our findings indicate a putative role of the polarized DBN-Rab8-dependent membrane trafficking for disease-specific autophagy, which is well aligned with astrocytes becoming phagocytic under pathological conditions[70]. The aspects of DBN deficiency identified in astrocytes in our study, are likely to exacerbate a mild CNS insult into major trauma with progressing neurodegeneration. Further dissection of the underlying mechanism could open new therapeutic avenues to treat CNS trauma and, possibly, degenerative conditions, which to date have dismal outcomes.

In conclusion, this study identifies an important function of injury-induced DBN in controlling damage containment in the CNS via membrane trafficking. As DBN and its downstream targets RAB8 and β1-integrin are broadly expressed, it is conceivable that this mechanism may be pivotal in other pathologies in the CNS as well as in different organ systems.

## Methods

**Ethical approval.** All animals were handled in accordance with the relevant national guidelines and regulations after ethical approval. Protocols were approved by the 'Landesamt für Gesundheit und Soziales' (LaGeSo; Regional Office for Health and Social Affairs) in Berlin, and animals are under the permit number G0189/14.

**Mouse strains.** $Dbn^{-/-}$ mice were described previously[12]. B6.CAMK:Cre/Dbn$^{fl/fl}$ mice were generated by crossing B6.Cg-Tg(Camk2a-cre)T29-1Stl/J (kindly provided by Dietmar Schmitz, Charité Berlin;[71]) with B6. Dbn$^{fl/fl}$ mice. BAC Aldh1L1 eGFP mice, used for initial stab wound experiments, were described previously[14]. Mice were housed in individually ventilated cages (IVCs). The cages contained wooden bedding material (SafeR Select, Safe), nestlets (Ancare), and a red, triangular plastic house (length: 12,5 cm, width: 11 cm, height: 6 cm; Tecniplast) or a plastic tunnel (length: 10 cm, diameter: 4,5 cm, in-house fabrication). The animals were maintained under standard conditions (room temperature: 22 ± 2 °C; relative humidity: 55 ± 10%) on a light:dark cycle of 12:12 h of artificial light (lights on from 6:00 a.m. to 6:00 p.m.). The mice were fed pelleted mouse diet ad libitum (Ssniff, V1534-000) and had free access to tap water at all times.

**Antibodies and reagents.** Antibodies and their used concentrations in Western blotting (WB), immunofluorescence (IF), and immunohistochemistry (IHC): please note that several lots of each antibody were used over the years. Mouse anti-DBN M2F6 (Enzo Lifesciences, ADI-NBA-110-E, IF & IHC 1:100; WB 1:1000), rabbit anti-GFAP (Synaptic Systems, 173 002, IF & IHC 1:1000), guinea pig anti-GFAP (Synaptic Systems, 173 004, IHC 1:400), guinea pig anti-S100β (Synaptic Systems, 287 003, IHC 1:200); rabbit anti-S100β (Atlas Antibodies, HPA015768, IHC 1:1000), mouse anti-alpha tubulin DM1a (Sigma-Aldrich, T6199, WB 1:5000), mouse anti-NeuN A60 (Millipore, MAB377, IHC 1:1000), rabbit anti-IBA1 (Wako, 019-19741, IHC 1:500), anti-MAP2 (Synaptic systems, 188 004, IHC 1:500), mouse anti-GAPDH 6C5 (Abcam, ab8245, Wb 1:5000), mouse anti-RAB8 (BD Biosciences, 610844, WB 1:1000), goat anti-pan RAB8 (Sicgen, AB3176-200, IF & IHC 1:100), rabbit anti-integrin beta1 (Cell Signaling, #4706 S, WB 1:1000), mouse anti-CD29 18/CD29 (BD Biosciences, WB 1:1000), rat anti-active integrin beta1/CD29 9EG7 (BD Pharmingen, 553715, IF 1:100), rabbit anti-integrin beta1 c-term (LSBio, LS-C413122, IF 1:100), rabbit anti-paxillin (Genetex, GTX125891, IF (1:250), rabbit anti-GST (Abcam, #9085-200 µl, WB 1:500). Horseradish peroxidase (HRP)-conjugated secondary antibodies (WB 1:5000): Goat Anti-Rabbit IgG Antibody (H + L) (VectorLabs, PI-1000), Horse Anti-Mouse IgG Antibody (H + L) (Vectorlabs, PI-2000); cross-absorbed secondary antibodies conjugated to cyanine or Alexa dyes were purchased from Dianova (IF & IHC 1:250): Donkey IgG anti-Mouse IgG (H + L)-Alexa Fluor 488 (Dianova, 715-545-150), Donkey IgG anti-Mouse IgG (H + L)-Alexa Fluor 647 (Dianova, 715-605-150); Donkey IgG anti-Goat IgG (H + L)-Alexa Fluor 488 (Dianova, 705-545-147), Donkey IgG anti-Guinea Pig IgG (H + L)-Cy3 (Dianova,706-165-148), Donkey IgG anti-Rabbit IgG (H + L)-Alexa Fluor 647 (Dianova, 711-605-152), Donkey IgG anti-Rabbit IgG (H + L)-Alexa Fluor 488 (Dianova, 711-545-152), Donkey IgG anti-Rabbit IgG (H + L)-Cy3 (Dianova, 711-165-152). Filamentous actin was labeled with ActiStain488, 555, or 670 phalloidin (Tebu Bio, PHDG1-A, PHDH1-A, or PHDN1-A, 1:250). DNA staining was carried out using Hoechst 33258 (Thermo Scientific, H3569, 1:10,000).

**Plasmids.** Lifeact-GFP was cloned into the pCDF backbone (SBI), after replacing copGFP with a suitable multiple cloning site via XbaI/SalI digest (Sequence of MCS: TCTAGAGCTAGCGCTACCGGTCGCCACCATGGGATGTACAGCGGC CGCGTCGAC). Accordingly, the CMV promoter was substituted with the astrocyte-specific gfaABC1 promoter, kindly provided by Michael Brenner (University of Alabama, US), using PCR (pGFAP-5:ATAGATATCAACATATCCTGG TGTGGAGTAGGG, pGFAP-3:ATAGCTAGCGCGAGCAGCGGAGGTGATGC GTC).

pEGFP-RAB8A was a gift from Daniel Gerlich (Addgene plasmid # 31803[72]. EGFP-Rab8a was cloned via PCR into the pCDF backbone for viral expression (pCDF-GFP-5:GACCTCCATAGAAGATTCTAGAGCTAGCATGGTGAGC AAGGGCGAGGAGCTGTTC, pCDF-RAB8A-3: GTAATCCAGAGGTTGATT GTCGACTCACACGAAGAACACATCGGAAAAAGCTGC).

mRuby- RAB8A was generated through replacement of EGFP in pCDF-eGFP-RAB8A with mRuby via NheI/BsrgI from pcDNA3-mRuby kindly provided by Nils Rademacher (Charité Universitätsmedizin Berlin, Germany). The lentiviral construct pCDF-paxillin-EGFP was subcloned from pEGFP-N3-paxillin (gift from Rick Horwitz;[73] Addgene plasmid #15233) into pCDF-GFAP:Lifeact-GFP via NheI/BsRGI after silent mutation of an internal BsRGI site (Paxillin-BsrGImut-s: GAGGGAGGAACACGTGTATAGCTTCCCAAACAA GCAG; Paxillin-BsRGImut-as: CTGCTTGTTTGGGAAGCTATACACGTGT TCCTCCTC).

Previously published pEYFP-N1-DBN E was co-transfected in rescue experiments in $Dbn^{-/-}$ astrocytes in conjunction with pCDF-mRuby-RAB8A[74]. pEYFP-N1 (Takara Bio Inc) was used as a corresponding negative control. pDEST15-hSlp4-a was purchased from Addgene (Plasmid#40046). All used primers and oligonucleotides are listed in the Supplementary information (Tab. S8).

**Cell culture and plasmid transfection.** Cortical astrocytes were isolated from WT or $Dbn^{-/-}$ mouse brains according to procedures for rats[75]. Cerebral cortices from P2 mice were isolated, mechanically dissociated in Hanks' Balanced Salt Solution (HBSS) and trypsinized (Life Technologies) for 15 min at 37 °C. Afterwards, trypsin (Invitrogen) was inhibited by triturating cells in complete Dulbecco's modified Eagle's medium (DMEM, Lonza) with 10% FBS. Cell suspensions were subsequently plated into T75 flasks, coated with collagen-I (0.025%, BD Biosciences) and 100 µg/ml poly-ornithine (Sigma). Microglia were erased by treating astrocytes cultures at high density for 90 min with 60 mM L-leucine-methylester (LME, Sigma) in complete medium. Astrocytes were cultivated to 90% of confluency and, subsequently, plated for imaging or biochemical experiments. Enriched microglia cultures were acquired by shaking T75 cm flasks with polygonal astrocytes at 150 rpm at 37 °C for 2 h prior to treatment with LME. Supernatant with floating microglia was plated on poly-ornithine-coated glass coverslips. Microglia cultures contained <5% astrocytes. Primary mixed cortical cultures were dissected from male and female embryonic day 16.5 WT or $Dbn^{-/-}$ mice[76]. Cortices were isolated and digested for 15 min with 10% Trypsin in HBSS (Life Technologies), washed with HBSS and triturated to single cells with glass pipets. Cells were plated on poly-ornithine (15 µg/ml)-coated coverslips and cultured in Neurobasal A medium (Life Technologies) containing 2% B27 (Life Technologies), 1% penicillin/streptomycin (Life Technologies), 100 µM β-mercaptoethanol (Applichem), and 1% GlutaMAX (Life Technologies). HEK293TN were obtained from BioCAT (Cat. no. LV900A1-GVO-SBI). All culturing materials were sterile and cell culture techniques were undertaken in Class II vertical laminar flow

cabinets (ThermoFisher Scientific). HEK293TN cells were not authenticated. Cultured HEK293TN cells were tested negative for mycoplasma. Astrocytes were transfected using TransIT LT1 (Mirus) according to the manufacturer's protocol.

**Lentivirus production and astrocyte infection.** Probes of fluorescent RAB8A and paxillin were expressed in astrocytes via lentiviral transduction. FIV particles were produced in the HEK293TN producer cell line (BioCat) and harvested analogously to HIV particles[77]. In all, 70% confluent astrocytes from WT and $Dbn^{-/-}$ mice on glass-bottom dishes were transduced 7 days before scratch wounding and live-cell imaging.

**RNA interference.** Astrocytes were transfected with siRNAs using Lipofectamine RNAiMax (Thermo Scientific) according to the manufacturer's instructions. The following published siRNA oligonucleotides were used (Sigma-Aldrich): siRab8a (GGAAUAAGUGUGAUGUGAA[78], siRab8b (GAAUGAUCCUGGGUAACAA[79], and control siRNA (AGGUAGUGUAAUCGCCUUGUU)[75], (Tab. S8).

**Scratch wound in vitro and live-cell imaging.** Confluent astrocytes were cultured in phenol red-free DMEM (Thermo Scientific) with 10% fetal bovine serum (FBS). Scratch wounds were performed by pulling a 200 µl-pipette tip through the confluent astrocyte layer as established previously:[80]. To subject the entire monolayer to the scratch injury, different scratch patterns were performed: for 18 mm coverslips, one scratch was performed vertically and one scratch was performed horizontally. For 4 well µ-slides dishes (IBIDI), one vertical scratch was performed. For 30 mm dishes, three horizontal and three vertical scratches were performed. Injured astrocytes were either biochemically analyzed via western blot or studied in live-cell imaging. Cell behavior was followed by live-cell imaging for 22 or 30 h using a Nikon Widefield with CCD camera, scanning stage, and environmental control chamber (OKO lab) and a ×40 objective (N.A. 0.7) with 1.5× intermediate magnification. Cells were imaged at 20-min intervals with composite large images (3 × 3 fields of view). Imaging start, unless specified otherwise, was 4 h after in vitro injury. Astrocytes in compiled movies were analyzed using kymographs in Fiji[81].

**Quantitation of membrane tubules.** Tubular membrane compartments were quantified based on a macro for Fiji, originally designed to analyze analogous structures in still images in heart muscle[82]. We extended the functionality of the original code by enabling analyses of z-stack image sequences acquired in live imaging experiments. The macro has been made available on github (see Code Availability).

**Pharmacological treatments.** In all, 100 nM Cytochalasin D (Merck Calbiochem, Cat. no. 250255), 100 µM CK-666 (Sigma-Aldrich, Cat. no: SML0006-5MG) or 25 µM SMIFH2 (Sigma-Aldrich, Cat. no. S4826-5MG) were pre-diluted in phenol red-free DMEM and added to GFP-RAB8A-expressing astrocytes after recording baseline for 30 min without treatment. Microtubule dynamics were imaged after incubating astrocytes for six hours in 1 µM SiR-tubulin (Spirochrome, cat#: SC002) in phenol red-free DMEM.

**Immunocytochemistry.** Cultured astrocytes were fixed using 3.7% formaldehyde in cytoskeleton-preservation buffer (25 mM HEPES, 60 mM PIPES, 10 mM EGTA, 2 mM MgCl$_2$, pH 7.4) for 20 min. After three washes with cytoskeleton-preservation buffer, cell permeabilization, while maintaining the utmost integrity of the cytoskeleton and endosomes, was achieved through 3 min incubation with 0.02% Triton X-100 in Phosphate-buffered saline (PBS). After three washes with PBS and brief incubation in 1% bovine serum albumin (BSA) in PBS, cells were incubated with primary antibodies diluted in PBS for 1 h at RT. After three washes with PBS and brief incubation in 1% BSA in PBS, cells were incubated in highly cross-absorbed secondary antibodies for another hour at RT. After three washes with PBS, cells were mounted in Mowiol. To visualize endogenous RAB8 tubules, PBS was generally replaced by the cytoskeleton-preservation buffer. Cells were permeabilized by incubation with 0.02% Triton X-100 in the cytoskeleton-preservation buffer for 30 min at RT. Wash steps were extended to 5 min each.

**Confocal microscopy and imaging processing.** Cells were imaged via multitrack mode on either Leica Sp8 (Leica) or Nikon A1Rsi+ (Nikon) confocal microscopes. Large image and multipoint scans were performed on Nikon A1 microscopes using an automated stage. Image processing were performed in FiJi and/or Imaris (Bitplane). To isolate and highlight DBN immunreactivity only in astrocytes, we merged GFP and GFAP in the CoLoc application of Imaris to create a mask. DBN signals within this mask resulted in "Astrocyte merge" by subtracting signals outside of astrocytes from the DBN channel. To categorize palisading astrocytes, we adapting protocols from previous publications[3,14]. We analyzed GFAP+ astrocytes in areas 300 µm around to the core lesion site. Polarized astrocytes with longer processes beyond the typical ~25 µm radius of non-polarized astrocytes were taken into account as "palisading". Most long processes were orientated perpendicularly to the core lesion. However, we detected occasionally astrocytes with long processes in diverting angles, which might be caused by collateral tissue damage. We included those cells in our quantification as well. "%palisading astrocytes" were

quantified as subset relative to the total number of GFAP+ astrocytes in the defined areas. Microglia morphometry was analyzed based on IBA1 immunoreactivity and the surface algorithm of Imaris. Kymographs were generated and analyzed via FiJi. Movies were compiled and annotated with Hitfilm Express 13 (FXhome Limited).

**Focal adhesion analysis.** To assess focal adhesion size, primary astrocytes were transduced with FIV lentivirus encoding for pCDF-GFAP:paxillin-EGFP. Seven days after transduction, medium was changed to phenol-free DMEM, and scratch injury was performed, followed by live-cell imaging for 22 h in 20 min intervals.

Acquired images were submitted to the Focal Adhesion Analysis Server[83] with the following specifications: Imaging frequency: 20 min; min. adhesion size 20, max. adhesion size 500; all further settings were left at default. The analyzed files obtained from the adhesion server were used to identify a region of interest at the leading edge of astrocytes extending into the scratch. Each Region of Interest contained at least 10 focal adhesions. Using the open source software INKSCAPE, the assigned number of every analyzed focal adhesion was identified and matched with the corresponding data (maximum adhesion size, mean adhesion size) from the excel-sheet as described on the focal adhesion analysis server website (https://faas.bme.unc.edu/results_understanding).

**Surface biotinylation and internalization assay.** Cultured astrocytes were incubated in 1 mg/ml cell-impermeant EZ-Link Sulfo-NHS-SS-Biotin (ThermoFisher) in ice-cold PBS pH 8 for 2 h. Any remaining unreacted biotin was removed by three washes with ice-cold PBS followed by 10 min of quenching with 100 mM glycine in ice-cold PBS. Cells were then split into four groups: (1) directed analyses of surface biotinylated proteins after labeling and quenching, (2) specificity control of surface biotinylation by removing biotin surface labeling with 50 mM 2-mercaptoethanesulfonic acid sodium salt (MESNA) in wash buffer (150 mM NaCl, 0.2% BSA, 20 mM Tris, pH 8.6), (3) pulse-chase analyses of internalized surface proteins after 15 min or (4) 30 min incubation at 37 °C in complete medium followed by MESNA-dependent removal of the surface-label. After three washes with wash buffer, samples were treated with 50 mM MESNA in wash buffer for 1 h on ice. Cell lysis of all samples was performed after two washes with ice-cold PBS by incubating cells for 20 min in ice-cold 1% Triton X-100 in PBS with protease inhibitor cocktails (Merck, Calbiochem set III, Cat. no. 539134), subsequent thorough scraping and sonication. Biotinylated proteins were isolated by incubating the samples with equal total protein amounts overnight at 4 °C and gentle agitation with 30 µl of Neutravidin-magnetic beads (Fisher Scientific, Cat. no. 11864143). The following day, beads were washed three times in ice-cold PBS + 1% Triton X-100 and twice with ice-cold PBS only. Samples were then subjected to sodium dodecyl sulfate polyacrylamide gel electrophoresis (SDS-PAGE) and WB probing with antibodies.

**Protein lysate preparation, SDS–PAGE, and WB.** Cultured astrocytes were washed once with cold PBS and lysed in cold radioimmunoprecipitation assay buffer, supplemented with protease inhibitors (Merck, Calbiochem set III, Cat. no. 539134). Cell lysates were centrifuged at $20,000 \times g$ and supernatant was transferred to a tube containing Roti load I SDS sample buffer. On average, 15–30 µg of protein was loaded on SDS–PAGE gel. Western blot analysis was performed by transferring proteins to nitrocellulose membranes using a wet blot tank system (Bio-Rad) for 2 h. The membranes were then blocked for 1 h at room temperature with 5% skim milk before incubating with primary antibodies overnight. Membranes were then washed 3 × 10 min in TBS-T and incubated with HRP-coupled secondary antibody for 1 hour, followed by 3 × 10 washes in TBS-T before detection. Membranes were imaged using the Fusion SL system from Vilber Lourmat. Quantification of band densities was performed using FIJI. The area of the band and the mean gray value were measured to obtain relative density. For relative quantifications, measurements were normalized to loading control.

**Antibody-feeding assay.** To visualize internalized active β1-integrin, astrocytes were first starved (DMEM without serum) for one hour and subsequently incubated with the 9EG7 antibody (BD Bioscience, 1:20 in DMEM) on ice for 1 h. After three washes with cold complete medium, cells were incubated for 30 min at 37 °C and 5% CO$_2$. Astrocytes were washed several times in PBS and then surface antibody was stripped with ice-cold acetic acid pH3 (0.5 M NaCl, 0.5% Acetic Acid in ddH2O) followed by several washes in PBS before fixation.

**In vivo stab wound.** Mice were anesthetized with a mixture of ketamine (100 mg/kg) and xylazine (10 mg/kg), and head-fixed on a stereotactic frame (Kopf Stereotax). Throughout the operation, body temperature was maintained at 36–37 °C, using a heating blanket. Full anesthesia of the mice was verified throughout the procedure by carefully checking breathing and reflexes using the pinching toe method. For surgery, an incision was created in the scalp and a small craniotomy was drilled above M1 motor cortex (bregma: −1 mm; lateral: 1 mm). An injection needle (Hamilton, gauge 33 was carefully inserted into the motor cortex and moved up and down three times (0.8 mm). The needle was removed and the scalp was sutured. Metamizol (5 mg/ml) was added to drinking water as analgesics until sacrifice. After surgery, mice were replaced in their cage, and kept warm during

wake–up and recovery on a heating plate at 37 °C. Stab wound outcomes were analyzed 7 or 30 days later by IHC and confocal microscopy.

**IHC and tissue clearance**. Seven or 30 days after stab wound injury, mice were sedated with isoflurane, perfused with 4% formaldehyde and sacrificed. Coronal sections (60 μm diameter) were obtained, permeabilized with 1% Triton X-100 in PBS and blocked in 5% BSA. In a first step, sections were labeled with IHC for GFAP to label reactive astrocytes and identify lesion sites with adjacent scars. GFAP-positive sections were stained for additional marker proteins and subsequently fixed in 4% formaldehyde for 1 h at 4 °C. Tissue clearance was obtained by incubating brain slices in ScaleA2 (4 M UREA, 10% (wt/vol) glycerol, 0.1% (wt/vol) Triton X-100 and 0.1× PBS) for 48 h. Transparent slices were mounted on glass slides using Mowiol with 4 M UREA and analyzed using a Nikon A1 confocal microscope with a ×40 objective (N.A. 1.3, working distance 240 μm) and automated scanning stage. For visualization of RAB8 tubules in vivo, animals were sacrificed as described above and perfused with 3.7% formaldehyde in a cytoskeleton-preservation buffer (25 mM HEPES, 60 mM PIPES, 10 mM EGTA, 2 mM MgCl$_2$, pH 7.4). Slices were permeabilized with 0.02% Triton X-100 overnight and stained as described above in cytoskeleton-preservation buffer. For visualization of RAB8 tubules in vivo, no tissue clearing was performed and cells were mounted on glass slides using Mowiol without urea.

**Histology**. Brain slices adhered on superfrost coverslips overnight. Slices were pre-incubating in a descending ethanol series (96%-, 90%-, 70%-, 50%- and 30% ethanol, each step for 2 min). Cresyl violet (acetate) Certistain (Sigma-Aldrich) working solution was freshly prepared in manufacturer's acetate buffer solution (pH3.6) and filtered prior to use. Then slices were stained in cresyl violet solution for 20 min. Subsequently, slices were incubated for 1 sec in 96% ethanol and washed briefly with 70% ethanol. Clearing of slices was performed by two subsequent incubations in xylene for 2 min. Stained slices were mounted in water-free DPX medium (Sigma-Aldrich).

**Purification of recombinant GST-hSLP4A and isolation of endogenous GTP-bound RAB8 from astrocytes**. Recombinant GST-hSLP4A was expressed in and purified from BL21 Rosetta DE3 *Eschechia coli* (Merck, Ca. no. 70954). In all, 5 ml of saturated pDEST15-hSlp4-a-transformed BL21 Rosetta DE3 *E. coli* culture were diluted in 500 ml 2× YT (Sigma-Aldrich, Cat. no. Y2377-250G) with Ampicillin (Sigma-Aldrich, Cat. no. A9518) under shaking (230 rpm) for 3 h at 37 °C. Protein expression was induced by 1 mM isopropyl β-d-1-thiogalactopyranoside (Sigma-Aldrich, Cat. no. I6758) and occurred overnight at room temperature under shaking. Bacteria were harvested by centrifugation (Ja10 rotor, 5000 rcf for 15 min at 4 °C) and stored at −80 °C. Purification of GST-hSLP4A was performed directly before the isolation of GTP-bound RAB8 from astrocytes: bacteria were resuspended in (50 mM HEPES, 250 mM NaCl, 10% glycerol plus 1% Triton X-100, supplemented with protease inhibitors (Merck, Calbiochem set III, Cat. No. 539134) and subsequently sonicated. Lysates were cleared by centrifugation (JA20 rotor, 25,402 × g, 25 min at 4 °C). Glutathione sepharose beads (Sigma-Aldrich, Cat. no. GE17-0756-01) were washed three times in HTG buffer (1% Triton X-100, 25 mm HEPES, 150 mM NaCl, 10% glycerol). Bacterial lysates were incubated with glutathione sepharose beads for 1 hour on a rotating wheel at 4 °C. Beads were washed three times with ice-cold lysis buffer (20 mM HEPES, 150 mM NaCl, 0.5% Triton X-100). Subsequently, beads were incubated with astrocyte lysate buffer supplemented with protease inhibitors (Merck, Calbiochem set III, Cat. no. 539134) for 1 hour at 4 °C on a rotating wheel. After four times of washing with lysis buffer with protease inhibitors, samples were subjected to SDS–PAGE and WB.

**Electron microscopy**. Brain slices, fixed, permeabilized, and labeled for GFAP, were cryoprotected stepwise in 0.1 M sodium phosphate buffer pH 7.4 (PB) supplemented with increasing concentrations of glycerol [10–20–30% (v/v)] and left overnight in 30% glycerol in PBS at 4 °C. The tissue was frozen by plunging into hexane (Carl Roth) at −70 °C. Samples were transferred into cold methanol (−90 °C) in a freeze-substitution chamber (Leica EM AFS). Methanol was replaced three times before the specimens were immersed overnight in anhydrous methanol at −90 °C, containing 2% (w/v) uranyl acetate. After rinsing several times with methanol, the temperature was gradually raised to −50 °C and left overnight at −50 °C. Tissue was then infiltrated with a mix of Lowicryl HM20 resin (Polysciences) and methanol (1:2; 1:1; 2:1, 1 h each) and left in pure resin overnight at −50 °C. Samples were transferred to flat embedding molds containing freshly prepared resin at −50 °C. UV polymerization was started at −50 °C (overnight) and then continued for 4 d at temperatures gradually increasing from −50 °C to −20 °C (24 h) and finally to +20 °C (24 h). Ultrathin sections (70 nm) were mounted on 200-mesh formvar-coated nickel grids (Plano). Images were acquired using a Zeiss EM 900 equipped with a digital camera (Proscan 1 K Slow-Scan CCD Camera).

**Statistics and reproducibility**. All statistical analyses were performed in Graph-Pad Prism software (Prism 7.0). Details on the statistical tests applied are provided within the figure legends. The data are reported as bar graphs displaying individual values and means ± SEM, as indicated in the figure legends, or as box-and-whiskers plots, ranging from minimum to maximum values. All data were tested for normality and accordingly subjected to parametric or non-parametric statistical analysis. Test results for normality distribution are reported in the figure legends only when data were not normally distributed. No experiments were excluded from the analyses.

**Reporting summary**. Further information on research design is available in the Nature Research Reporting Summary linked to this article.

## Data availability

All data supporting the findings of this study are provided within the paper and its supplementary information. All additional information will be made available upon reasonable request to the authors. Cartoons and schemes depicted in this manuscript were created by J.S. and K.M. with BioRender.com Source data are provided with this paper.

## Code availability

The script to quantify tubules has been uploaded to Github and can be downloaded via the following link: https://github.com/jschiweck/TubuleMacro.git

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

## Acknowledgements

We thank Magdalena Götz and Sofia Grade (LMU Munich) for the provision of slices from the BAC Aldh1L1:GFP reporter mice, expert opinion, and critical feedback on the manuscript. We thank Heike Heilmann, Kerstin Schlawe, Kristin Lehmann, and Beate Diemar for excellent technical assistance. We thank the Advanced Medical BioImaging Core Facility (AMBIO) and the NeuroCure multi-user Microscopy Core Facility for usage of microscopes. We thank Dietmar Schmitz for providing the B6.Cg-Tg(Camk2a-cre)T29-1Stl/J mouse line. Funding was provided by the DFG (SFB 958 and 'Sachbeihilfe' - Project 285933818, NeuroCure EXC257); Sonnenfeld Stiftung (J.S.).

## Author contributions

J.S., K.M., A.M.W., M.O., and J.L. performed the experiments. J.S and K.M. analyzed the experiments. J.S. developed the automated script for tubule quantification. K.M., B.J.E., and I.V. supervised experiments. K.M. and B.J.E. designed the study. K.M., B.J.E., and J.S. wrote the manuscript.

## Funding

## Competing interests

The authors declare no competing interests.
