## [Peer Review File · Nature Communications]

Reviewers' Comments:

Reviewer #1:

Remarks to the Author:

The manuscript by Schiweck et al reports the role of actin-binding protein Drebrin (DBN) in regulating the formation of RAB8-positive tubular endosomes in astrocytes, thus impacting reactive astrogliosis and astrocyte scar formation in a stab model of brain injury. The authors provided extensive cell biological evidence to support the function of DBN in shifting actin dynamics from ARP2/3-dependent arrays to microtubule-compatible scaffolds such that RAB8 membrane tubules may be formed. DBN knockout also exhibited defects in the internalization of β 1-integrin in vitro and the accumulation of intracellular membrane in astrocyte processes and endfeet at the ultrastructural level in vivo. Overall, the evidence is strong in supporting authors' hypothesis, and the study provides novel insight on the function of DBN and in understanding the mechanism of astrocyte reactivity and scar formation. The significance is high and the topic is of broad interest.

There are some weaknesses, however, as written. The following points are meant to improve the study:

- 1) The term "glial scar" is somewhat out of date, as it is a vague term. Most researchers have switched to the term "astrocyte scar". The astrocyte scar is only part of the injury scar (the other major component being the fibrotic scar).
- 2) "In the mouse brain, DBN was distributed in distinct puncta in proximity to MAP2-positive dendrites, but not in S100 β + astrocytes (Figures S1B and 1C)." This is hard to tell from the image. DBN appears to be everywhere except where S100 β + is and is low where MAP2 is. Also, why not showing side-by-side comparison between uninjured and injured mice for all markers and analyses so that this phenotype can be better understood? In Figure S1B, GFAP is not shown in blue. How is "Astrocyte DBN merge" generated?
- 3) Figure legend (or text) should explain all the arrows in figure panels (e.g., Fig. 1A, but many others).
- 4) Does the word "palisading" suggest the orientation of the processes being perpendicular to the injury border? Please clarify either way. Figure 2A, how is % palisading astrocytes (e.g. 45%) quantified?
- 5) Figure 2D, The lack of GFAP and altered pattern of SOX9 at the injury site are interesting. There should be a baseline level of GFAP signals throughout the cord just as in uninjured mice. Figure 2D seems to suggest that this baseline level GFAP is not present. This needs to be clarified. The same applies to SOX9. It will help to show some adjacent tissue (more normal tissue) to better understand this phenotype. To substantiate this phenotype, what about the pattern of vimentin (another marker for reactive astrocytes) and Aldh1l1 (immunoreactivity or the GFP reporter expression)?
- 6) The authors stress that both the formation and maintenance of the astrocyte scar are disrupted? How are these two processes distinguished?
- 7) It is a bit puzzling that the authors used a *Dbnf1/fl:CAMK-Cre* mouse line to exclude the role of neuronal DBN, but conducted the entire study with a germline knockout and did not use astrocyte specific DBN knockout to pinpoint the astrocyte role. Did the authors try either GFAP-CreERT2 (which could be problematic here but one does not need a lot of Cre expression to induce gene deletion) or another astrocyte specific Cre line in combination with *Dbnf/f* mice?
- 8) How can one be sure that the tubular structures in Fig. 3D, F, G are the same types of structures (in vitro and in vivo)? Why not showing a DBN knockout equivalent image of Fig. 3G to verify that these tubules are no longer prevalent in the mutants?
- 9) Figure 4A rescue experiment, DBN-YFP and mRuby-RAB9A seem co-localized. Will be interesting to discuss.
- 10) Page 15, how are total β 1-integrin vs. active β 1-integrin detected differently?
- 11) For discussion, how much does astrocyte motility or migration come into play to explain the in vivo phenotype?

Minor:

- The use of "ad hoc" is odd and can be confusing.
- "During this stab injury" should be "After this stab injury" (page 4).
- Page 13, awkward sentence: "...but they did show no signs to form tubules."
- Figure S6B, images reversed between ipsilateral and contralateral.

Reviewer #2:

Remarks to the Author:

This paper reports a combination of in vivo and in vitro experiments providing multiple types of evidence that the actin-binding protein, debrin, is essential for astrocytes to form neuroprotective scar borders around areas of CNS damage. Genetic deletion of debrin leads to failure of astrocyte scar border formation after CNS injury in vivo. Detailed in vitro experiments show that debrin regulates actin networks to organize into scaffolds crucial for membrane trafficking and coordinated cellular polarization and palisade-like outgrowth of astrocyte processes essential for scar border formation. The design of experiments is thoughtful and incisive. The experimental evidence provided is detailed, rigorous and compelling. Experiments are properly controlled and have sufficient replicates to have statistical power. The figures presented are of high quality. I could not find any major technical issues or concerns. The findings are not over-interpreted and the discussion is well balanced and the findings are put into context. The findings are important because they begin to provide detailed information on intracellular molecular events that underlie and are crucial for the formation of neuroprotective scar borders by astrocytes, which occurs in response to all forms of damage to CNS tissue. Loss or dysfunction of debrin activities has the potential to exacerbate various forms of CNS disorders. Although I have no concerns regarding technical aspects of the study, I have some minor suggestions regarding the background information provided the paper.

Specific comment:

1) At the end of the first paragraph of the introduction, the authors say that "Glial scars are anatomically well described, but the molecular details controlling astrocyte reactivity are still poorly understood." This statement is not entirely correct. Dozens of molecules have been identified that play important roles in controlling astrocyte reactivity and formation of functional scar borders by astrocytes and there are many, many papers on this topic as reviewed in Sofroniew 2015 Nat Rev Neuroci <https://www.nature.com/articles/nrn3898>. Acknowledging this does not detract from the findings under review, which provide important new information. A more accurate statement would be "Glial scars are well described anatomically and with respect to certain molecular signaling events (ref <https://www.nature.com/articles/nrn3898>), but little is known about molecular events controlling organization of actin scaffolds, cellular polarization and palisade-like outgrowth of astrocyte processes crucial for scar border formation" or some phrasing along those lines that the authors might prefer.

Reviewer #3:

Remarks to the Author:

formation. The authors first show up-regulation of Drebrin only after injury in astrocytes, followed by convincing data on the deficits after stab wound injury when this up-regulation can not occur in the knock-out condition. Astrocyte polarization is severely reduced and their GFAP levels at scar stages are gone. To which extent this is indeed reduced scar formation or aggravated damage as indicated by loss of NeuN and cytoplasmic translocation of Sox9 remains to be better determined. Notably, deletion of Drebrin in neurons does not result in such a phenotype. Most importantly, however, the authors then proceed to work out the mechanisms of Drebrin

function in reactive astrocytes using in vitro models in combination with some in vivo verification. They show that Drebrin regulates formation of Rab8 tubular endosomes that influence e.g. β 1 integrin localization to the focal adhesions. Interestingly, low concentrations of cytochalasin to depolymerize the actin cytoskeleton or blocking Arp2/3, rescues appearance of the tubules. Importantly, the authors then use ultrastructural analysis to demonstrate vacuoles in Drebrin^{-/-} reactive astrocytes after injury, demonstrating accumulation of membrane filled vesicles also in vivo. These are very interesting and novel findings that highlight the key relevance of the endosomal transport system in reactive astrocytes after injury. I have only a few suggestions to further improve the quality of this exciting manuscript.

Suggestions:

- 1) It's not clear if the loss of GFAP⁺ astrocytes at 30 days post injury represents reduced scar formation or simply defects in the astrocytes. To check this, the authors could perform cresyl violet staining and quantify the size of the lesion/wound/scar, they could stain for collagens to label the scar-associated ECM, and stain for CD45⁺ monocytes that also often accumulate in the scar.
- 2) I am intrigued by the cytoplasmic translocation of NeuN in Figure 2C and wonder if this is a cell autonomous effect. Do the authors still see this in the injured CAMK-Cre mice? The cytoplasmic translocation of Sox9 probably does not occur in these mice then. Could the authors discuss how Drebrin would affect the nuclear-cytoplasm shuttling?
- 3) Figure 1A – please specify in the legends what the arrows are supposed to indicate.
- 4) Please specify in the legend if the panels show a single optical section or a confocal stack. For Figure 1C an orthogonal projection would be good to evaluate if Drebrin is really within the astrocyte processes.
- 5) It would be interesting to know if Drebrin is up-regulated transcriptionally or only posttranscriptionally – please check in published data sets from the Barres lab (Zamanian et al. 2012 and thereafter), the Götz lab (Sirko et al., 2015) and the Hol lab (Kamphuis et al., 2015) also to discuss if this role of Drebrin may be restricted to certain injury models.

Reviewer's Comments

Reviewer #1 (Remarks to the Author):

The manuscript by Schiweck et al reports the role of actin-binding protein Drebrin (DBN) in regulating the formation of RAB8-positive tubular endosomes in astrocytes, thus impacting reactive astrogliosis and astrocyte scar formation in a stab model of brain injury. The authors provided extensive cell biological evidence to support the function of DBN in shifting actin dynamics from ARP2/3-dependent arrays to microtubule-compatible scaffolds such that RAB8 membrane tubules may be formed. DBN knockout also exhibited defects in the internalization of β 1-integrin in vitro and the accumulation of intracellular membrane in astrocyte processes and endfeet at the ultrastructural level in vivo. Overall, the evidence is strong in supporting authors' hypothesis, and the study provides novel insight on the function of DBN and in understanding the mechanism of astrocyte reactivity and scar formation. The significance is high and the topic is of broad interest.

There are some weaknesses, however, as written. The following points are meant to improve the study:

1) The term "glial scar" is somewhat out of date, as it is a vague term. Most researchers have switched to the term "astrocyte scar". The astrocyte scar is only part of the injury scar (the other major component being the fibrotic scar).

2) "In the mouse brain, DBN was distributed in distinct puncta in proximity to MAP2-positive dendrites, but not in S100 β + astrocytes (Figures S1B and 1C)." This is hard to tell from the image. DBN appears to be everywhere except where S100 β + is and is low where MAP2 is. Also, why not showing side-by-side comparison between uninjured and injured mice for all markers and analyses so that this phenotype can be better understood? In Figure S1B, GFAP is not shown in blue. How is "Astrocyte DBN merge" generated?

3) Figure legend (or text) should explain all the arrows in figure panels (e.g., Fig. 1A, but many others).

4) Does the word "palisading" suggest the orientation of the processes being perpendicular to the injury border? Please clarify either way. Figure 2A, how is % palisading astrocytes (e.g. 45%) quantified?

5) Figure 2D, The lack of GFAP and altered pattern of SOX9 at the injury site are interesting. There should be a baseline level of GFAP signals throughout the cord just as in uninjured mice. Figure 2D seems to suggest that this baseline level GFAP is not present. This needs to be clarified. The same applies to SOX9. It will help to show some adjacent tissue (more normal tissue) to better understand this phenotype. To substantiate this phenotype, what about the pattern of vimentin (another marker for reactive astrocytes) and Aldh111 (immunoreactivity or the GFP reporter expression)?

6) The authors stress that both the formation and maintenance of the astrocyte scar are disrupted? How are these two processes distinguished?

7) It is a bit puzzling that the authors used a *Dbn1^{fl/fl}:CAMK-Cre* mouse line to exclude the role of neuronal DBN, but conducted the entire study with a germline knockout and did not use astrocyte specific DBN knockout to pinpoint the astrocyte role. Did the authors try either GFAP-CreERT2 (which could be problematic here but one does not need a lot of Cre expression to induce gene deletion) or another astrocyte specific Cre line in combination with *Dbn1^{fl/fl}* mice?

8) How can one be sure that the tubular structures in Fig. 3D, F, G are the same types of structures (in vitro and in vivo)? Why not showing a DBN knockout equivalent image of Fig. 3G to verify that these tubules are no longer prevalent in the mutants?

9) Figure 4A rescue experiment, DBN-YFP and mRuby-RAB9A seem co-localized. Will be interesting to discuss.

10) Page 15, how are total β 1-integrin vs. active β 1-integrin detected differently?

11) For discussion, how much does astrocyte motility or migration come into play to explain the in vivo phenotype?

Minor:

- The use of “ad hoc” is odd and can be confusing.
- “During this stab injury” should be “After this stab injury” (page 4).
- Page 13, awkward sentence: “...but they did show no signs to form tubules.”
- Figure S6B, images reversed between ipsilateral and contralateral.

Reviewer #2 (Remarks to the Author):

This paper reports a combination of in vivo and in vitro experiments providing multiple types of evidence that the actin-binding protein, debrin, is essential for astrocytes to form neuroprotective scar borders around areas of CNS damage. Genetic deletion of debrin leads to failure of astrocyte scar border formation after CNS injury in vivo. Detailed in vitro experiments show that debrin regulates actin networks to organize into scaffolds crucial for membrane trafficking and coordinated cellular polarization and palisade-like outgrowth of astrocyte processes essential for scar border formation. The design of experiments is thoughtful and incisive. The experimental evidence provided is detailed, rigorous and compelling. Experiments are properly controlled and have sufficient replicates to have statistical power. The figures presented are of high quality. I could not find any major technical issues or concerns. The findings are not over-interpreted and the discussion is well balanced and the findings are put into context. The findings are important because they begin to provide detailed information on intracellular molecular events that underlie and are crucial for the formation of neuroprotective scar borders by astrocytes, which occurs in response to all forms of damage to CNS tissue. Loss or dysfunction of debrin activities has the potential to exacerbate various forms of CNS disorders. Although I have no concerns regarding technical aspects of the study, I have some minor suggestions regarding the background information provided the paper.

Specific comment:

1) At the end of the first paragraph of the introduction, the authors say that “Glial scars are anatomically well described, but the molecular details controlling astrocyte reactivity are still poorly understood.” This statement is not entirely correct. Dozens of molecules have been identified that play important roles in controlling astrocyte reactivity and formation of functional scar borders by astrocytes and there are many, many papers on this topic as reviewed in

Sofroniew 2015 Nat Rev Neuroci <https://www.nature.com/articles/nrn3898> . Acknowledging this does not detract from the findings under review, which provide important new information. A more accurate statement would be “Glial scars are well described anatomically and with respect to certain molecular signaling events (ref <https://www.nature.com/articles/nrn3898>), but little is known about molecular events controlling organization of actin scaffolds, cellular polarization and palisade-like outgrowth of astrocyte processes crucial for scar border formation” or some phrasing along those lines that the authors might prefer.

Reviewer #3 (Remarks to the Author):

formation. The authors first show up-regulation of Drebrin only after injury in astrocytes, followed by convincing data on the deficits after stab wound injury when this up-regulation can not occur in the knock-out condition. Astrocyte polarization is severely reduced and their GFAP levels at scar stages are gone. To which extent this is indeed reduced scar formation or aggravated damage as indicated by loss of NeuN and cytoplasmic translocation of Sox9 remains to be better determined. Notably, deletion of Drebrin in neurons does not result in such a phenotype.

Most importantly, however, the authors then proceed to work out the mechanisms of Drebrin function in reactive astrocytes using in vitro models in combination with some in vivo verification. They show that Drebrin regulates formation of Rab8 tubular endosomes that influence e.g. b1integrin localization to the focal adhesions. Interestingly, low concentrations of cytochalasin to depolymerize the actin cytoskeleton or blocking Arp2/3, rescues appearance of the tubules. Importantly, the authors then use ultrastructural analysis to demonstrate vacuoles in Drebrin-/reactive astrocytes after injury, demonstrating accumulation of membrane filled vesicles also in vivo. These are very interesting and novel findings that highlight the key relevance of the endosomal transport system in reactive astrocytes after injury. I have only a few suggestions to further improve the quality of this exciting manuscript.

Suggestions:

- 1) Its not clear if the loss of GFAP+ astrocytes at 30days post injury represents reduced scar formation or simply defects in the astrocytes. To check this, the authors could perform cresylviolet staining and quantify the size of the lesion/wound/scar, they could stain for collagens to label the scar-associated ECM, and stain for CD45+ monocytes that also often accumulate in the scar.
- 2) I am intrigued by the cytoplasmic translocation of NeuN in Figure 2C and wonder if this is a cell autonomous effect. Do the authors still see this in the injured CAMK-Cre mice? The cytoplasmic translocalization of Sox9 probably does not occur in these mice then. Could the authors discuss how Drebrin would affect the nuclear-cytoplasm shuttling?
- 3) Figure 1A – please specify in the legends what the arrows are supposed to indicate.
- 4) Please specify in the legend if the panels show a single optical section or a confocal stack. For Figure 1C an orthogonal projection would be good to evaluate if Drebrin is really within the astrocyte processes.
- 5) It would be interesting to know if Drebrin is up-regulated transcriptionally or only posttranscriptionally – please check in published data sets from the Barres lab (Zamanian et al. 2012 and thereafter), the Götz lab (Sirko et al., 2015) and the Hol lab (Kamphuis et al., 2015) also to discuss if this role of Drebrin may be restricted to certain injury models.

Detailed responses to reviewer comments.

Reviewer comments are shown in black and our response in blue.

Response to Reviewer #1

Reviewer #1 (Remarks to the Author):

The manuscript by Schiweck et al reports the role of actin-binding protein Drebrin (DBN) in regulating the formation of RAB8-positive tubular endosomes in astrocytes, thus impacting reactive astrogliosis and astrocyte scar formation in a stab model of brain injury. The authors provided extensive cell biological evidence to support the function of DBN in shifting actin dynamics from ARP2/3-dependent arrays to microtubule-compatible scaffolds such that RAB8 membrane tubules may be formed. DBN knockout also exhibited defects in the internalization of β 1-integrin in vitro and the accumulation of intracellular membrane in astrocyte processes and endfeet at the ultrastructural level in vivo. Overall, the evidence is strong in supporting authors' hypothesis, and the study provides novel insight on the function of DBN and in understanding the mechanism of astrocyte reactivity and scar formation. The significance is high and the topic is of broad interest.

We thank the referee for the positive and helpful comments, and for stating that our study is of high significance, broad interest and provides novel and mechanistic insight on astrocyte reactivity and scar formation.

Our responses to the specific comments are as follows:

1) The term “glial scar” is somewhat out of date, as it is a vague term. Most researchers have switched to the term “astrocyte scar”. The astrocyte scar is only part of the injury scar (the other major component being the fibrotic scar).

We replaced the term “glial” to “astrocyte”, where applicable.

2) “In the mouse brain, DBN was distributed in distinct puncta in proximity to MAP2-positive dendrites, but not in S100 β + astrocytes (Figures S1B and 1C).” This is hard to tell from the image. DBN appears to be everywhere except where S100 β + is and is low where MAP2 is. Also, why not showing side-by-side comparison between uninjured and injured mice for all markers and analyses so that this phenotype can be better understood?

We added a close-up image of Figure 1C to Supplementary Figure 1 to show that the distinct DBN+ puncta are dendritic spines emerging from MAP2+-dendrites. The *in vivo* pattern is thus in line with Figure 1A and previous publications ^{1, 2, 3}.

We used different markers in Figure 1C for the following reasons: S100 β is a cytoplasmic and membrane-bound protein and labels the complex morphology of astrocytes to great extent ⁴. We used this marker as counterstain to show that astrocytes are overall devoid of DBN under normal conditions. In our stab wound model, we decided not to use S100 β , as reactive astrocytes secrete this protein ⁵. Instead we used mice expressing GFP under control of the *Aldh1l1* promoter. Cytoplasmic GFP labels astrocyte processes to lesser extent than S100 β but it is robustly expressed in quiescent and reactive astrocytes ⁶⁻⁸. To illustrate the key findings concisely, we would like to keep the compact version of Figure 1 in the main text. However, we

added new images to Figure S1, which show the absence of DBN from quiescent GFP+/GFAP-astrocytes in the uninjured tissue.

In Figure S1B, GFAP is not shown in blue.

We apologize for the mistake on the GFAP labeling in Figure S1B, which was a remainder from previous manuscript versions. The caption of the composite image in Figure S1B is corrected in the new version.

How is “Astrocyte DBN merge” generated?

‘Astrocyte DBN’ was generated in IMARIS by using the combined GFP and GFAP channels to generate a mask, which in turn was used to isolate and highlight DBN immunoreactivity specifically in astrocytes. We added a corresponding paragraph to the ‘Methods’ section (page 31, line 1-4).

3) Figure legend (or text) should explain all the arrows in figure panels (e.g., Fig. 1A, but many others).

We added the missing information on the arrows to the corresponding figure legends.

4) Does the word “palisading” suggest the orientation of the processes being perpendicular to the injury border? Please clarify either way. Figure 2A, how is % palisading astrocytes (e.g. 45%) quantified?

We categorized palisading astrocytes by adapting protocols from previous publications^{9,10}. We analyzed GFAP+ astrocytes in an area 300 μm adjacent to the core lesion site. Astrocytes with longer processes beyond the typical (approx.) 25-μm radius of non-polarized astrocytes were regarded as ‘palisading’. Most long processes were orientated perpendicularly. However, we occasionally detected astrocytes with long processes with diverting angles, which might be caused by collateral tissue damage. We included those cells in our quantification as well. ‘%palisading astrocytes’ were quantified as subset relative to the total number of GFAP+ astrocytes in the defined areas. This information is now included in the ‘Methods’ section of our revised manuscript (page 31, lines 5-11).

5) Figure 2D, The lack of GFAP and altered pattern of SOX9 at the injury site are interesting. There should be a baseline level of GFAP signals throughout the cord just as in uninjured mice. Figure 2D seems to suggest that this baseline level GFAP is not present. This needs to be clarified. The same applies to SOX9. It will help to show some adjacent tissue (more normal tissue) to better understand this phenotype. To substantiate this phenotype, what about the pattern of vimentin (another marker for reactive astrocytes) and Aldh1l1 (immunoreactivity or the GFP reporter expression)?

To answer the reviewer’s questions, we added overview images of GFAP, vimentin and SOX9-labeled cortex to the new Supplementary Figures 2 and 3. To show the distribution of Aldh1l1, we added close-up images of WT and DBN^{-/-} astrocytes to Supplementary Figure 3.

We would like to emphasize, that, in contrast to astroglia in brain white matter and spinal cord, the majority of cortical mouse astrocytes downregulate GFAP to great extent^{11,12}. They appear thus as 'GFAP negative' in our and other immunohistochemistry experiments¹³. Pathological changes, like stab wounding, induce the re-expression of GFAP, which thereby serves as specific marker for reactive grey matter astrocytes. Vimentin mirrors the expression profile of GFAP in astrocytes: Cortical WT astrocytes do not express detectable levels of vimentin under healthy conditions but upregulate it locally in scars after stab injury. In contrast, DBN^{-/-} astrocytes lose vimentin immunoreactivity within 30 days post injury. Endogenous ALDH1L1 protein is detectable in quiescent and reactive astrocytes of WT and DBN^{-/-} astrocytes. However, the ALDH1L1 immunoreactivity in GFAP-/cytoplasmic SOX9+ cells in injured DBN^{-/-} is substantially lower than in corresponding WT astrocytes (Figure S3B). Sox9 labels in the uninjured brain tissue of all studied genotypes the nuclei of astrocytes (Figure S3A), as published previously¹⁴.

6) The authors stress that both the formation and maintenance of the astrocyte scar are disrupted? How are these two processes distinguished?

The reviewer's questions showed us that our original statement in the abstract was indeed misleading. It was our intention to refer to 'formation', on the one hand, to the defective polarization and outgrowth of DBN^{-/-} astrocytes detected seven days post injury. These cells express at this time point GFAP and were therefore (still) 'reactive'. On the other hand, we aimed to emphasize the inability of DBN^{-/-} astrocytes to sustain their astrogliosis program, which becomes obvious by no-longer detectable marker proteins for 'reactive astrocytes' and escalating neurodegeneration. We have clarified our statement by re-phrasing as following: '..., which is essential for the scar formation and maintenance of astrocyte reactivity *in vivo*.' (page 2, lines 5-6).

7) It is a bit puzzling that the authors used a Dbnf1/fl:CAMK-Cre mouse line to exclude the role of neuronal DBN, but conducted the entire study with a germline knockout and did not use astrocyte specific DBN knockout to pinpoint the astrocyte role. Did the authors try either GFAP-CreERT2 (which could be problematic here but one does not need a lot of Cre expression to induce gene deletion) or another astrocyte specific Cre line in combination with Dbnf/f mice?

We principally agree with the reviewer; the combinatorial approach of a germline and a neuron-specific knockout to study astrocyte-specific effects may not necessarily appear as 'straightforward'. We decided not to use astrocyte-specific knockout mouse lines for the following reasons: As the reviewer already mentioned, a specific but comprehensive drebrin knockout in astrocytes by CreERT2 and tamoxifen administrations is very difficult. According to previous publications and personal information from experts in the field, recombination efficiencies can be rather variable between experiments, brain regions and gene loci¹⁵⁻¹⁷. Transgenic mouse lines expressing constitutively active cre recombinase under control of the truncated human GFAP, endogenous mouse GFAP or ALDH1L1 promoter show also gene ablations in some neurons and/or oligodendrocytes¹⁸⁻²⁰. In view of these limitations, the restricted expression pattern of drebrin and the germline knockout exhibiting phenotypes only under conditions of stress and injury, we consider our approach as suitable to analyze the role of drebrin in reactive astrocytes.

8) How can one be sure that the tubular structures in Fig. 3D, F, G are the same types of

structures (in vitro and in vivo)? Why not showing a DBN knockout equivalent image of Fig. 3G to verify that these tubules are no longer prevalent in the mutants?

We thank the reviewer for this very constructive suggestion. We added to Figure 3F and 3G images of DBN^{-/-} astrocytes in injured mixed cultures and at stab wound sites (7 DPI), as the reviewer requested. Analogous to cultured cells depicted in Figure 3 and Figure 5, as well as the ultrastructural analyses shown in Figure 6, we discovered accumulating vesicular- and vacuole-like structures in DBN^{-/-} astrocytes *in vivo*. The RAB8⁺ accumulations were particularly frequent in soma of DBN^{-/-} astrocytes and origins of very short tubular structures. In addition, we also observed in direct proximity to GFAP⁺ processes RAB8⁺ structures (Figure S6D), which resembled membrane cisterns in cultured DBN^{-/-} astrocytes, shown in Figure 5. Besides these structures, RAB8 was mostly dispersed in processes of DBN^{-/-} astrocytes *in vivo*. These results further support our model that DBN is essential for the formation of Rab8⁺ tubules in reactive astrocytes.

9) Figure 4A rescue experiment, DBN-YFP and mRuby-RAB9A seem co-localized. Will be interesting to discuss.

We discuss in the revised paragraph ‘DBN antagonizes ARP2/3-dependent actin dynamics’ (page 22, lines 19-24) the co-localization of DBN and RAB8A and the possibility of DBN directly associating with and/or creating scaffolds around RAB8A tubules.

10) Page 15, how are total β 1-integrin vs. active β 1-integrin detected differently?

We used different antibodies to detect β 1-integrin, in dependence or independence of its activation state. The well-established 9EG7 antibody labels active β 1-integrin on living and fixed cells²¹. To study total β 1-integrin by western blotting, we used pan β 1-integrin antibodies. However, direct comparisons between active and total β 1-integrin, by these means, are not possible: On the one hand, the 9EG7 epitope is not preserved under denaturing conditions. In contrast to experiments with human cells and human tissue, no β 1-integrin antibody is currently available that faithfully recognizes total β 1-integrin in murine cells. We are therefore restricted in detecting distinct integrin subpopulations, such as active β 1-integrin and β 1-integrin in endosomal compartments.

11) For discussion, how much does astrocyte motility or migration come into play to explain the *in vivo* phenotype?

We thank the reviewer for the constructive suggestion. The local outgrowth of astrocytic processes into injury sites is crucial for scar formation and damage containment, as astrocytes from adjoining areas do not migrate into lesions^{10,22,23}. We explain this aspect of CNS injury in the revised paragraph ‘DBN function in brain astrocytes’ (page 24, lines 18-23).

Minor:

- The use of “ad hoc” is odd and can be confusing.

We have replaced “ad hoc” with “immediate” to avoid any potential confusion or misunderstandings.

- “During this stab injury” should be “After this stab injury” (page 4).

We re-phrased the sentence according to the reviewer’s suggestion.

- Page 13, awkward sentence: “...but they did show no signs to form tubules.”

We clarified our description on GFP-RAB8a vesicles in *Dbn*^{-/-} astrocytes by shortening the corresponding sentence to “... but they did not form tubules”

- Figure S6B, images reversed between ipsilateral and contralateral.

We thank the reviewer for bringing this error to our attention. The images are now correctly organized as described in the text.

Response to Reviewer #2

Reviewer #2 (Remarks to the Author):

This paper reports a combination of in vivo and in vitro experiments providing multiple types of evidence that the actin-binding protein, debrin, is essential for astrocytes to form neuroprotective scar borders around areas of CNS damage. Genetic deletion of debrin leads to failure of astrocyte scar border formation after CNS injury in vivo. Detailed in vitro experiments show that debrin regulates actin networks to organize into scaffolds crucial for membrane trafficking and coordinated cellular polarization and palisade-like outgrowth of astrocyte processes essential for scar border formation. The design of experiments is thoughtful and incisive. The experimental evidence provided is detailed, rigorous and compelling. Experiments are properly controlled and have sufficient replicates to have statistical power. The figures presented are of high quality. I could not find any major technical issues or concerns. The findings are not over-interpreted and the discussion is well balanced and the findings are put into context. The findings are important because they begin to provide detailed information on intracellular molecular events that underlie and are crucial for the formation of neuroprotective scar borders by astrocytes, which occurs in response to all forms of damage to CNS tissue. Loss or dysfunction of debrin activities has the potential to exacerbate various forms of CNS disorders. Although I have no concerns regarding technical aspects of the study, I have some minor suggestions regarding the background information provided the paper.

We thank the reviewer for the helpful comments and for stating that our study is important by providing novel detailed information on the molecular mechanisms of neuroprotective scar formation. The reviewer also highlights that our experimental design is thoughtful and incisive, and our findings are supported by detailed, rigorous and compelling evidence, which are properly controlled and put into context by a well-balanced discussion.

Specific comment:

1) At the end of the first paragraph of the introduction, the authors say that “Glial scars are anatomically well described, but the molecular details controlling astrocyte reactivity are still poorly understood.” This statement is not entirely correct. Dozens of molecules have been identified that play important roles in controlling astrocyte reactivity and formation of functional scar borders by astrocytes and there are many, many papers on this topic as reviewed in Sofroniew 2015 Nat Rev Neuroci <https://www.nature.com/articles/nrn3898> . Acknowledging this does not detract from the findings under review, which provide important new information. A more accurate statement would be “Glial scars are well described anatomically and with respect to certain molecular signaling events (ref <https://www.nature.com/articles/nrn3898>), but little is known about molecular events controlling organization of actin scaffolds, cellular polarization and palisade-like outgrowth of astrocyte processes crucial for scar border formation” or some phrasing along those lines that the authors might prefer.

Our response to the specific comment is as follows:

We thank the reviewer for his expert advice regarding the already identified molecular triggers and signaling molecules involved in astrogliosis responses. It wasn't our intention to neglect this wealth of knowledge. Instead, our aim was to emphasize the lack of insight on downstream effectors like the cytoskeleton and membrane trafficking. We believe that we have accomplished our aim and acknowledged previous research by re-phrasing the respective sentence along the reviewer's suggestions and citing the appropriate literature.

Response to Reviewer #3

Reviewer #3 (Remarks to the Author):

formation. The authors first show up-regulation of Drebrin only after injury in astrocytes, followed by convincing data on the deficits after stab wound injury when this up-regulation can not occur in the knock-out condition. Astrocyte polarization is severely reduced and their GFAP levels at scar stages are gone. To which extent this is indeed reduced scar formation or aggravated damage as indicated by loss of NeuN and cytoplasmic translocation of Sox9 remains to be better determined. Notably, deletion of Drebrin in neurons does not result in such a phenotype.

Most importantly, however, the authors then proceed to work out the mechanisms of Drebrin function in reactive astrocytes using in vitro models in combination with some in vivo verification. They show that Drebrin regulates formation of Rab8 tubular endosomes that influence e.g. β 1 integrin localization to the focal adhesions. Interestingly, low concentrations of cytochalasin to depolymerize the actin cytoskeleton or blocking Arp2/3, rescues appearance of the tubules. Importantly, the authors then use ultrastructural analysis to demonstrate vacuoles in Drebrin-/- reactive astrocytes after injury, demonstrating accumulation of membrane filled vesicles also in vivo. These are very interesting and novel findings that highlight the key relevance of the endosomal transport system in reactive astrocytes after injury. I have only a few suggestions to further improve the quality of this exciting manuscript.

We thank the referee for the positive and helpful comments, and for stating that our manuscript is exciting and provides very interesting and novel findings.

Our responses to the specific comments are as follows:

1) Its not clear if the loss of GFAP+ astrocytes at 30 days post injury represents reduced scar formation or simply defects in the astrocytes. To check this, the authors could perform cresylviolett staining and quantify the size of the lesion/wound/scar, they could stain for collagens to label the scar-associated ECM, and stain for CD45+ monocytes that also often accumulate in the scar.

We thank the reviewer for these excellent suggestions. Cresyl violet stainings in $DBN^{-/-}$ mice show numerous cells with intensely labeled nuclei along and peripheral to the core lesion. These cells are likely to be glia cells. Their localization correlates with the distribution of cytoplasmic Sox9+ astrocytes, shown in Figure 2.

Principally, the Cresyl violet labeling supports our hypothesis of $DBN^{-/-}$ astrocytes being still present but unable to maintain their reactivity. However, our immunohistochemical and histological analyzes do not completely exclude the possibility of limited astrocyte cell death. A definite answer to this open question could be achieved by sophisticated experiments involving astrocytes, which are labeled under healthy conditions *in vivo* and then traced before and after injury through cranial windows and two-photon imaging¹⁰. We will pursue this avenue in our future research. However, in the present manuscript we added the data obtained by cresyl violet stainings to Supplementary figure S4.

In addition, cresyl violet labelings in $DBN^{-/-}$ brains also show areas peripheral to the stab wounds largely devoid of Nissl body+ neurons. These findings confirm the results; we have shown in Figure 2 by NeuN antibody labeling. The combination of immunohistochemistry and histology strongly supports the relevance of our findings on drebrin deficiency turning a 'harmless' deemed local injury into an exacerbating neurodegenerative condition in the long run. Accordingly, we added these results to Supplementary figure S4.

We also stained ECM components such as CSPG and collagen IV 30 days post stab injury. In $Dbn^{-/-}$ brains, the results between experiments and animals were inconsistent. While these ECM components were in some $DBN^{-/-}$ animals comparable to their WT brains, we found in another subset of $DBN^{-/-}$ animals a potential reduction in CSPG and collagen IV. As ECM components in brain injuries change over time, we would like to study in the future these markers in more animals and at different time points (3, 5 and 120 DPI). To avoid at this point a misinterpretation in this aspect of brain injury, we would like to exclude these results from our manuscript.

As the reviewer suggested, we also analyzed the abundance of CD45+ monocytes. We detected CD45+ monocytes in both stab wounded WT and $Dbn^{-/-}$ mice, where they were confided in comparable numbers to the core lesions. We have inserted these results in Figure S4B.

2) I am intrigued by the cytoplasmic translocation of NeuN in Figure 2C and wonder if this is a cell autonomous effect. Do the authors still see this in the injured CAMK-Cre mice? The cytoplasmic translocalization of Sox9 probably does not occur in these mice then. Could the authors discuss how Drebrin would affect the nuclear-cytoplasm shuttling?

We included in our revised manuscript analyzes on NeuN and Sox9 in stab wounds of CAMK-cre mice. Neither NeuN nor Sox9 translocate at injury sites of CAMK-cre-mice. These new data are now shown in Supplementary figures S3A and S4C. In contrast to other actin binding

proteins, such as profilin²⁴, we never observed a nuclear localization or nuclear-cytoplasmic shuttling of drebrin. We therefore interpret the effects of drebrin loss on NeuN and Sox9 localizations as indirect.

The translocation of NeuN has been described in few publications as early sign of neuronal stress^{7,25-27}, while most publications use NeuN loss as readout for neurodegeneration^{9,23}. It is conceivable that the initial NeuN translocation is mostly overlooked in studies with disease and injury models, which are considerably more severe than our approach.

The neuronal response to injuries in *Dbn*^{-/-} mice could indeed be cell autonomous, caused by detrimental cues leaking through the defective astrocyte scar. However, cell non-autonomous mechanisms are also conceivable: The obvious trafficking defects in *Dbn*^{-/-} astrocytes may affect their communication with neurons and other cells. Thereby, *Dbn*^{-/-} astrocytes may not be able to react appropriately and provide the support that neurons require to survive in pathological settings.

As DBN is not localized in astrocyte nuclei at any point, we consider defects in astrocyte surface receptor trafficking as plausible explanation for the SOX9 translocation and lost astrocyte reactivity. This idea is supported by, for instance, findings in spinal cord injuries, where antibody injections interfered with surface b1-integrin trafficking and functions in astrocytes and led to GFAP downregulation and reduced astrogliosis²⁸.

3) Figure 1A – please specify in the legends what the arrows are supposed to indicate.

We added the missing information on the arrows to the legend of figure 1A.

4) Please specify in the legend if the panels show a single optical section or a confocal stack. For Figure 1C an orthogonal projection would be good to evaluate if Drebrin is really within the astrocyte processes.

We added the missing information to the corresponding figure legends. The image in Figure 1C is a single optical section, while the corresponding panel in Figure S1B shows the stab wound as a confocal stack. To further proof the localization of DBN in astrocyte processes, we added an animated orthogonal projection of Figure S1B as new Movie 1 to the Supplement.

5) It would be interesting to know if Drebrin is up-regulated transcriptionally or only posttranscriptionally – please check in published data sets from the Barres lab (Zamanian et al. 2012 and thereafter), the Götz lab (Sirko et al., 2015) and the Hol lab (Kamphuis et al., 2015) also to discuss if this role of Drebrin may be restricted to certain injury models.

We thank the reviewer for the excellent advice to explore astrocyte-specific transcriptome data bases for disease-related changes in DBN mRNA. In line with our findings, focal-penetrating traumatic injuries like cortical stab wounds and spinal cord injuries (SCI) lead to significant increases in DBN transcript levels in reactive astrocytes (Stab wound 7 DPI - change DBN mRNA reactive vs. control astrocytes: logfc 1.74²⁹; SCI - change DBN mRNA injured astrocytes vs. uninjured astrocytes: logfc 1.37³⁰). Pathologies without mechanical tissue damage like the MCAO stroke model or systemic CNS inflammation by LPS injections cause less pronounced changes in astrocyte DBN mRNA levels (DBN change 7d post MCAO vs. 7d sham control: logfc 0.299; DBN change LPS vs. saline control: logfc 0.302³¹). As the reviewer suggested, we discuss these data in context with our findings in the revised manuscript.

In an APP^{swe}/PS1^{dE9} double-transgenic AD mouse model, astrocyte DBN mRNA is reduced in astrocytes (change DBN mRNA WT vs. APP^{swe}/PS1^{dE9} astrocytes: log₂fc -2.26³²). This finding is interesting and relevant to our future research. However, this transcriptome analysis relies on one late time point, when the AD-like pathology is widely progressed. It is conceivable that DBN transcript levels are different at earlier disease stages. For this reason as well as the current controversial debate in the field on reactive astrogliosis in AD, we decided not to include the findings by Kamphuis et al. in the discussion of our revised manuscript.

In addition to DBN mRNA changes, posttranslational modification may also contribute to increased DBN protein levels in reactive astrocytes. This would be in line with our previous study describing ATM-dependent phosphorylation as mechanism to increased DBN protein lifetime in neurons during oxidative stress². Accordingly, we discussed this possibility in our revised manuscript.

General comments

We adapted the diagrams in the revised manuscript according to the Nature Communications format. Experiments with sample sizes of n<10 were shown as bar graphs displaying individual values and means ± SEM. Experiments with sample sizes of n> 10 were displayed as box-and-whiskers plots, ranging from minimum to maximum values. We improved statistical testing where appropriate. The minor changes in statistical tests did not alter the outcomes of our study.

References

1. Shirao, T., Inoue, H. K., Kano, Y. & Obata, K. Localization of a developmentally regulated neuron-specific protein S54 in dendrites as revealed by immunoelectron microscopy. *Brain Res.* **413**, 374–378 (1987).
2. Kreis, P. *et al.* ATM phosphorylation of the actin-binding protein drebrin controls oxidation stress-resistance in mammalian neurons and *C. elegans*. *Nat. Commun.* **10**, 486 (2019).
3. Dombroski, T. C. D. *et al.* Drebrin expression patterns in patients with refractory temporal lobe epilepsy and hippocampal sclerosis. *Epilepsia* **61**, 1581–1594 (2020).
4. S100B in neuropathologic states: The CRP of the brain? - Sen - 2007 - Journal of Neuroscience Research - Wiley Online Library. <https://onlinelibrary.wiley.com/doi/abs/10.1002/jnr.21211>.
5. Michetti, F. *et al.* The S100B story: from biomarker to active factor in neural injury. *J. Neurochem.* **148**, 168–187 (2019).
6. Benediktsson, A. M., Schachtele, S. J., Green, S. H. & Dailey, M. E. Ballistic labeling and dynamic imaging of astrocytes in organotypic hippocampal slice cultures. *J. Neurosci. Methods* **141**, 41–53 (2005).
7. Shandra, O. *et al.* Repetitive Diffuse Mild Traumatic Brain Injury Causes an Atypical Astrocyte Response and Spontaneous Recurrent Seizures. *J. Neurosci.* **39**, 1944–1963 (2019).
8. Mattugini, N. *et al.* Inducing Different Neuronal Subtypes from Astrocytes in the Injured Mouse Cerebral Cortex. *Neuron* **103**, 1086-1095.e5 (2019).

9. Faulkner, J. R. Reactive Astrocytes Protect Tissue and Preserve Function after Spinal Cord Injury. *J. Neurosci.* **24**, 2143–2155 (2004).
10. Bardehle, S. *et al.* Live imaging of astrocyte responses to acute injury reveals selective juxtavascular proliferation. *Nat. Neurosci.* **16**, 580–586 (2013).
11. Walz, W. Controversy surrounding the existence of discrete functional classes of astrocytes in adult gray matter. *Glia* **31**, 95–103 (2000).
12. Kimelberg, H. K. The problem of astrocyte identity. *Neurochem. Int.* **45**, 191–202 (2004).
13. Miyake, T., Okada, M. & Kitamura, T. Reactive proliferation of astrocytes studied by immunohistochemistry for proliferating cell nuclear antigen. *Brain Res.* **590**, 300–302 (1992).
14. Sun, W. *et al.* SOX9 Is an Astrocyte-Specific Nuclear Marker in the Adult Brain Outside the Neurogenic Regions. *J. Neurosci. Off. J. Soc. Neurosci.* **37**, 4493–4507 (2017).
15. Slezak, M. *et al.* Transgenic mice for conditional gene manipulation in astroglial cells. *Glia* **55**, 1565–1576 (2007).
16. Jahn, H. M. *et al.* Refined protocols of tamoxifen injection for inducible DNA recombination in mouse astroglia. *Sci. Rep.* **8**, 5913 (2018).
17. Park, Y. M., Chun, H., Shin, J.-I. & Lee, C. J. Astrocyte Specificity and Coverage of hGFAP-CreERT2 [Tg(GFAP-Cre/ERT2)13Kdmc] Mouse Line in Various Brain Regions. *Exp. Neurobiol.* **27**, 508–525 (2018).
18. Zhuo, L. *et al.* hGFAP-cre transgenic mice for manipulation of glial and neuronal function in vivo. *Genes. N. Y. N 2000* **31**, 85–94 (2001).
19. Garcia, A. D. R., Doan, N. B., Imura, T., Bush, T. G. & Sofroniew, M. V. GFAP-expressing progenitors are the principal source of constitutive neurogenesis in adult mouse forebrain. *Nat. Neurosci.* **7**, 1233–1241 (2004).
20. Tien, A.-C. *et al.* Regulated temporal-spatial astrocyte precursor cell proliferation involves BRAF signalling in mammalian spinal cord. *Dev. Camb. Engl.* **139**, 2477–2487 (2012).
21. Lenter, M. *et al.* A monoclonal antibody against an activation epitope on mouse integrin chain beta 1 blocks adhesion of lymphocytes to the endothelial integrin alpha 6 beta 1. *Proc. Natl. Acad. Sci. U. S. A.* **90**, 9051–9055 (1993).
22. Tsai, H.-H. *et al.* Regional astrocyte allocation regulates CNS synaptogenesis and repair. *Science* **337**, 358–362 (2012).
23. Wanner, I. B. *et al.* Glial scar borders are formed by newly proliferated, elongated astrocytes that interact to corral inflammatory and fibrotic cells via STAT3-dependent mechanisms after spinal cord injury. *J. Neurosci. Off. J. Soc. Neurosci.* **33**, 12870–12886 (2013).
24. Stüven, T., Hartmann, E. & Görlich, D. Exportin 6: a novel nuclear export receptor that is specific for profilin.actin complexes. *EMBO J.* **22**, 5928–5940 (2003).
25. Lucas, C.-H., Calvez, M., Babu, R. & Brown, A. Altered subcellular localization of the NeuN/Rbfox3 RNA splicing factor in HIV-associated neurocognitive disorders (HAND). *Neurosci. Lett.* **558**, 97–102 (2014).

26. Wiley, C. A. *et al.* Ultrastructure of Diaschisis Lesions after Traumatic Brain Injury. *J. Neurotrauma* **33**, 1866–1882 (2016).
27. Wang, J. *et al.* Comparison of different quantification methods to determine hippocampal damage after cerebral ischemia. *J. Neurosci. Methods* **240**, 67–76 (2015).
28. Hara, M. *et al.* Interaction of reactive astrocytes with type I collagen induces astrocytic scar formation through the integrin-N-cadherin pathway after spinal cord injury. *Nat. Med.* **23**, 818–828 (2017).
29. Sirko, S. *et al.* Astrocyte reactivity after brain injury-: The role of galectins 1 and 3. *Glia* **63**, 2340–2361 (2015).
30. Anderson, M. A. *et al.* Astrocyte scar formation aids central nervous system axon regeneration. *Nature* **532**, 195–200 (2016).
31. Zamanian, J. L. *et al.* Genomic analysis of reactive astrogliosis. *J. Neurosci. Off. J. Soc. Neurosci.* **32**, 6391–6410 (2012).
32. Kamphuis, W. *et al.* GFAP and vimentin deficiency alters gene expression in astrocytes and microglia in wild-type mice and changes the transcriptional response of reactive glia in mouse model for Alzheimer’s disease. *Glia* **63**, 1036–1056 (2015).

Reviewers' Comments:

Reviewer #1:

Remarks to the Author:

The authors' response to this reviewer's critiques is thorough, with reason and care. While it will be worth investigating further with inducible astrocyte-specific Drebrin gene deletion in future, the current manuscript has sufficient advance for a broad readership.

Reviewer #2:

Remarks to the Author:

The authors have appropriately dealt with my concerns and as far as I can tell have also addressed the concerns of the other reviewers. I have no additional comments or concerns. I continue to find the paper important and of high interest.

Reviewer #3:

Remarks to the Author:

The authors have fully addressed all my previous concerns and this beautiful work is now ready for publication. It describes a novel mechanism of maintaining astrocyte activation regulating aspects of scar formation of great interest for the readers of Nature Communications. I fully support its publication now.